# Drosophila models uncover substrate channeling effects on phospholipids and sphingolipids in peroxisomal biogenesis disorders

Michael F. Wangler[1,2]*, Yu-Hsin Chao[1,2¤], Mary Roth[3], Ruth Welti[3], James A. McNew[4]

1 Molecular and Human Genetics, Baylor College of Medicine, Houston, Texas, United States of America, 2 Jan and Dan Duncan Neurological Research Institute, Texas Children's Hospital, Houston, Texas, United States of America, 3 Kansas Lipidomics Research Center, Division of Biology, Kansas State University, Manhattan, Kansas, United States of America, 4 Department of BioSciences, Rice University, Houston, Texas, United States of America

¤ Present address: Senior Research Associate, Frontier Medicines, Boston.
* michael.wangler@bcm.edu

## Abstract

Peroxisomal Biogenesis Disorders Zellweger Spectrum (PBD-ZSD) disorders are a group of autosomal recessive defects in peroxisome formation that produce a multi-systemic disease presenting at birth or in childhood. Well documented clinical biomarkers such as elevated very long chain fatty acids (VLCFA) are key biochemical diagnostic findings in these conditions. Additional, secondary biochemical alterations such as elevated very long chain lysophosphatidylcholines are allowing newborn screening for peroxisomal disease. In addition, a more widespread impact on metabolism and lipids is increasingly being documented by metabolomic and lipidomic studies. Here we utilize *Drosophila* models of *pex2* and *pex16* as well as human plasma from individuals with *PEX1* mutations. We identify phospholipid abnormalities in *Drosophila* larvae and brain characterized by differences in the quantities of phosphatidylcholine (PC) and phosphatidylethanolamines (PE) with long chain lengths and reduced levels of intermediate chain lengths. For diacylglycerol (DAG), the precursor of PE and PC through the Kennedy pathway, the intermediate chain lengths are increased suggesting an imbalance between DAGs and PE and PC that suggests the two acyl chain pools are not in equilibrium. Altered acyl chain lengths are also observed in PE ceramides in the fly models. Interestingly, plasma from human subjects exhibit phospholipid alterations similar to the fly model. Moreover, human plasma shows reduced levels of sphingomyelin with 18 and 22 carbon lengths but normal levels of C24. Our results suggest that peroxisomal biogenesis defects alter shuttling of the acyl chains of multiple phospholipid and ceramide lipid classes. In contrast, DAG species with intermediate fatty acids are actually more abundant in PBD. These data suggest an imbalance between *de novo* synthesis of PC and PE through the Kennedy pathway and remodeling of existing PC and PE through the

**Data availability statement:** All relevant data are within the paper and its Supporting Information files.

**Funding:** This work was supported by the National Institute for Neurological Disorders and Stroke 5R01NS107733 to MFW. The lipid analyses described in this work were performed at the Kansas Lipidomics Research Center Analytical Laboratory. Instrument acquisition and lipidomics method development were supported by the National Science Foundation (including support from the Major Research Instrumentation program; most recent award DBI-1726527), K-IDeA Networks of Biomedical Research Excellence (INBRE) of National Institute of Health (P20GM103418), USDA National Institute of Food and Agriculture (Hatch/Multi-State project 1013013), and Kansas State University.

**Competing interests:** The authors have declared that no competing interests exist.

Lands cycle. This imbalance is likely due to overabundance of very long acyl chains in PBD and a subsequent imbalance due to substrate channeling effects. Given the fundamental role of phospholipid and sphingolipids in nervous system functions, these observations suggest PBD-ZSD are diseases characterized by widespread cell membrane lipid abnormalities.

## Introduction

Peroxisomal biogenesis is an evolutionarily conserved process that allows the cell to generate functional peroxisomes. Genetic mutations in the peroxisomal biogenesis machinery, affecting proteins which are encoded by the *PEX* genes in humans, lead to global biochemical defects that reflect lack of peroxisomes [1–3]. The lack of functional peroxisomes leads to a number of metabolic changes which relate directly to peroxisomal biochemistry such as elevations in very long chain fatty acids (VLCFAs), reduced plasmalogens, and increased phytanic and pipecolic acid [1,4]. Recently, newborn screening for peroxisomal disease, particularly X-linked Adrenoleukodystrophy has been initiated in multiple countries based on secondary biochemical alterations in peroxisomal dysfunction impacting lysophosphatidylcholines [2,5,6]. This screening ultimately depends on an imbalance in the length of acyl chains observed when peroxisomal fatty acid metabolism is altered or absent, stemming from the defect in peroxisomal beta-oxidation and leading to an over-abundance of very long acyl chains.

Metabolomic and lipidomic studies on peroxisomal disorders have documented the metabolic consequences of general peroxisomal defects and uncovered additional abnormalities in peroxisomal disorders beyond the classical peroxisomal biochemical markers [7–10]. These studies have demonstrated that a generalized defect in peroxisomal biochemistry is not restricted to affecting peroxisomal pathways but also produces a myriad of downstream secondary biochemical changes. Of note, the phospholipid compositions of fibroblast cells from patients with PBD-ZSD are altered and display an increas in phospholipid species containing very long-chain fatty acids and in addition and increased number of unsaturations in PBD [8]. Indeed, altered phospholipids have been shown to correlate with other biomarkers of peroxisomal disease and show potential for identification of new biomarkers [11]. In our previous metabolomic studies we also observed reduced plasma sphingomyelin levels in PBD-ZSD [9].

Phosphatidylethanolamine (PE) and phosphatidylcholine (PC) are the most abundant phospholipids and their synthesis is mediated in part by the Kennedy pathway which allows *de novo* synthesis of phospholipids from diacylglycerols (DAG) [12]. The biochemical steps for PE and PC synthesis do not occur within peroxisomes and so the effects on these lipids are examples of secondary alterations in metabolites that are not primarily peroxisomal in etiology. When considering these secondary metabolic consequences of PBD there are several considerations. Some of the secondary metabolic abnormalities could stem from a primary excess or deficiency of a substrate that comes from peroxisomal metabolism such as excess very long chain

fatty acid or deficient plasmalogens. In addition, regulation of overall cellular metabolism could be affected in PBD which could lead to secondary metabolic changes. In order to explore the full extent of the metabolic consequences of PBD one approach is to leverage animal models of these disorders and to perform comprehensive metabolic or lipidomic assessments. We have used *Drosophila* models such as *pex2* and *pex16* as well as *pex3* to fully catalogue the impact of peroxisomal biogenesis defects on metabolism [13,14]. Here we extend these analyses using a thorough phospholipid analysis [15,16] including PC and PE and their precursor DAGs in addition to ceramides and PE ceramides and sphingomyelin.

## Materials and methods

### Fly husbandry

All flies were maintained at room temperature (21°C) and except where otherwise noted experiments were conducted at room temperature. The *pex2¹*, and *pex2²* lines were derived from imprecise excision of (*w[1118]*; P{w[+mC] = EPg}*pex-2*[HP35039]/TM3, Sb[1], see below) these were then backcrossed 5 generations with *y w*: FRT80B and studied as:

*y w*; FRT80B
*y w*; FRT80B- *pex2¹*
y w; FRT80B- *pex2²*
*w[1118]*;PBac{y[+mDint2]w[+mC] = 53M21}VK00037;FRT80B- *pex2²*
w[1118];PBac{y[+mDint2]w[+mC] = 53M21}VK00037;FRT80B- *pex2¹*

Except where otherwise indicated the 5 strains above each crossed to a genomic deficiency uncovering *pex2* locus *w1118*; Df(3L)BSC376/TM6C,*Sb1 cu1* are labeled as *pex2* Control, *pex2², pex2¹, pex2²* Rescue, and *pex2¹* Rescue, respectively.

The *y w*: *pex16¹* line [41] was obtained from Kenji Matsuno, derived from:

*y 1 w67c23*; P {GSV6}*pex16*GS14106/TM3, Sb1 Ser1. The *y w*: *pex16^EY* strain was obtained from Bloomington Stock center

*y[1] w[67c23]*; P{*w* [+mC] y[+mDint2] = EPgy2}*Pex16*[EY05323].
*y w*; FRT80B
y w; pex161
y[1] w[67c23]; P{w[+mC] y[+mDint2] = EPgy2}pex16[EY05323]
w[1118];PBac{y[+mDint2]w[+mC] = 115M13}VK00037; *pex16¹*
w[1118];PBac{y[+mDint2]w[+mC] = 115M13}VK00037;*Pex16[*EY05323].

Except where otherwise indicated, the 5 strains above each crossed to a genomic deficiency uncovering the *pex16* locus *w1118*; Df(3L)BSC563/TM6C,*cu1 Sb1* are labeled as *pex16* Control, *p*ex16¹ pex16^EY pex16¹ Rescue and pex16^EY Rescue respectively.

### Human samples

All subjects were recruited to an institutional review board–approved study, "The Biochemical and Cell Biology of Peroxisomal Disorders Study," at Baylor College of Medicine (H-32837) which was open from 05/15/2013–01/25/2019 or "Translational Models of Neurological Disease Study" (H-44779) open from 12/04/2020 to present [10,17,18]. All subject caregivers/parents provided informed consent, including consent to publish patient photos. All participants were examined, and samples were collected under the same conditions. Ascertainment was based on the presence of molecularly confirmed mutations in the PEX genes and/or biochemical confirmation of a defect in peroxisomal biogenesis. Blood samples were collected by a trained phlebotomist and immediately plasma was isolated by ultracentrifugation. Each plasma sample was divided into aliquots and frozen at −80 ° C.

## Lipid extraction protocol

Lipids were extracted for Drosophila larvae and heads. The tissue was homogenized in aqueous solution, then 1 part chloroform and 2 parts methanol were added. The samples were shaken and additional 1 part chloroform and 1 part water were added. These were then centrifuged at low speed and the lower layer was removed with glass pipettes, the process of shaking, centrifuging and removing the lower layer was repeated three times. All the lower layers were then combined and washed in 1 M KCl and then washed with water. These lower layers were then evaporated completely under a nitrogen stream and shipped for analysis.

## Phospholipid and diacylglycerol analysis

*Drosophila* phospholipids and diacylglycerols were analyzed by direct-infusion into an electrospray ionization triple quadrupole mass spectrometer (Applied Biosystems 4000 Q-trap mass spectrometer, Sciex, Framingham, MA, US), as described [19]. The plasma phospholipids (3 µL plasma per sample) were analyzed similarly using a Waters Xevo TQS mass spectrometer (Waters Corp., Milford, MA, US). Preparation of plasma, internal standard information, and details are presented s[20] . Mass spectrometry parameters on the Xevo TQS were source temperature, 150°C; desolvation temperature, 250°C; cone gas flow, 150 L h$^{-1}$; desolvation gas flow, 650 L h$^{-1}$; collision gas flow, 0.14 mL min$^{-1}$; nebulizer gas, 7 Bar; LM 1 resolution, 3.2; HM 1 resolution, 15.5. Scan and data processing parameters are shown in Table S1.

## Statistical comparisons

Unless otherwise specified the analysis shown are results from student's T-test with p value <0.05 indicated by *, p value < 0.01 indicated by **, and p value <0.001 indicated by ***

## Results

### Drosophila pex2 and pex16 mutant lines

*Drosophila pex2* and *pex16* null mutants have been previously studied as peroxisomal loss-of-function models in flies and we utilized these mutant and genomic rescue controls in this study [13]. For *pex16* we utilize a deletion [21] and an EY insertion P-element line [22]. For pex2 both alleles utilized are considered null alleles that delete the gene. In the case of *pex16*, the *pex16* deletion allele is also considered a null, although the EY insertion preserves the coding sequence and has some attenuated phenotypes suggesting it is a hypomorph. In order to study *pex16* (**Fig 1A**) and *pex2* (**Fig 1B**) mutants, we utilize genomic rescue lines and controls as described [13]. Because we had previously used untargeted metabolomics and observed phospholipid abnormalities (Fig S1), we pursued lipidomic analysis of the *Drosophila pex2* and *pex16* mutants in larvae as well as adult heads (primarily Drosophila adult brain) (**Fig 1C**).

### Drosophila pex mutant larvae exhibit altered chain lengths of phosphatidylcholine and phosphatidylethanolamine

First, we examined the *pex* mutant at the *Drosophila* larval stage and focused on phospholipid analysis (Table S2). We observed that the relative amounts (in mol %) of phosphatidylethanolamine (PE) (**Fig 2A**), and phosphatidylcholine (PC) (**Fig 2B**) either exhibited minor differences or are equivalent between *pex2* mutant and rescue and *pex16* mutant and rescue. This was consistent with our previous study using untargeted metabolomic methods showing no difference in quantity for PC and PE [13]. However, while the total of all sub-types appears unchanged in *pex2* and *pex16* mutants, specific sub-types are altered and there is a clear pattern of these alterations related to the carbon chain length. Both PE 30:1 (**Fig 2C**) and PC 30:1 (**Fig 2D**) are reduced in *pex2* mutant compared to rescue and *pex16* mutant compared to rescue, although the difference was not statistically significant for *pex2* for PE 30:1. Note that in this nomenclature for PC and PE, the 30:1 means that the two acyl chains have a total of 30 carbons and 1 unsaturation total. Across the spectrum of phospholipids measured, all the acyl groups are likely between C12 and C22 and would be considered long chain fatty acids or acyl

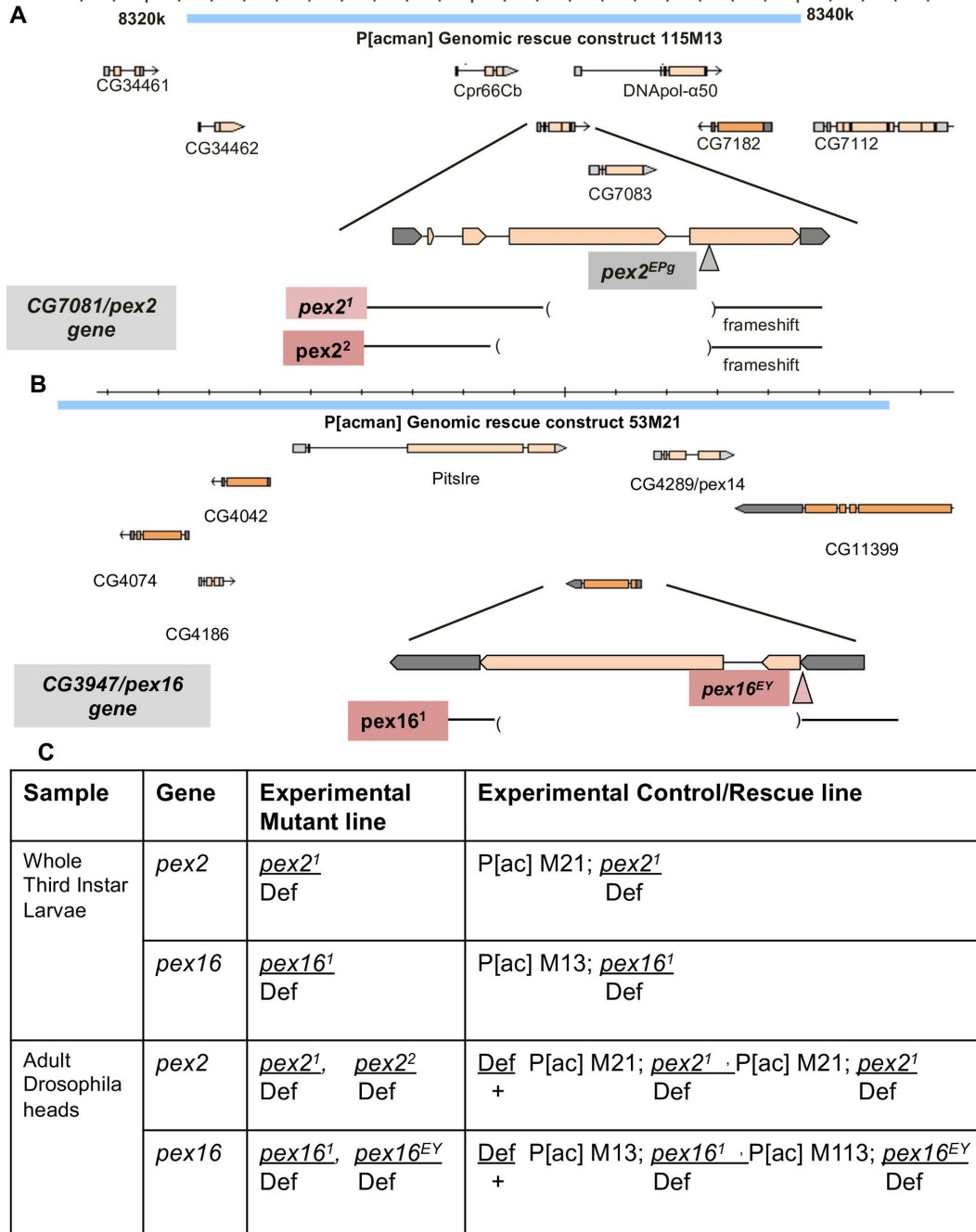

**Fig 1. Genotypes of Drosophila peroxisomal biogenesis mutant lines used in the lipidomic study.** A. The *pex2* gene (former CG7081) in *Drosophila melanogaster* is on 3L and is depicted on the chromosomal region. The *pex2¹*, and *pex2²* lines are deletion alleles. The genomic rescue construct line is derived from 115M13 clone and rescues the entire region (blue bar). B. The *pex16* gene (former CG3947) in *Drosophila melanogaster* is on 3L and is depicted on the chromosomal region. The *y w: pe16¹* line is depicted as a deletion allele (c/o Kenji Matsuno). The *y w: pex16^EY* strain is an EY insertion allele in the 5'UTR of the gene. The genomic rescue construct line is derived from the 53M21 clone (blue bar).C. The *Drosophila* experiments analyzed in this study were conducted on whole third instar larvae and then another set of experiments on adult *Drosophila* heads (brain and cuticle). The genotypes for each of these experiments include analysis of the *pex2¹* and the *pe16¹* allele were studied in comparison to genomics rescue lines. For the adult head experiments two alleles each for *pex2* and *pex16* were studied in comparison to both control and rescue lines. The boxes show labels for these lines, for full genotypes please refer to the methods section.

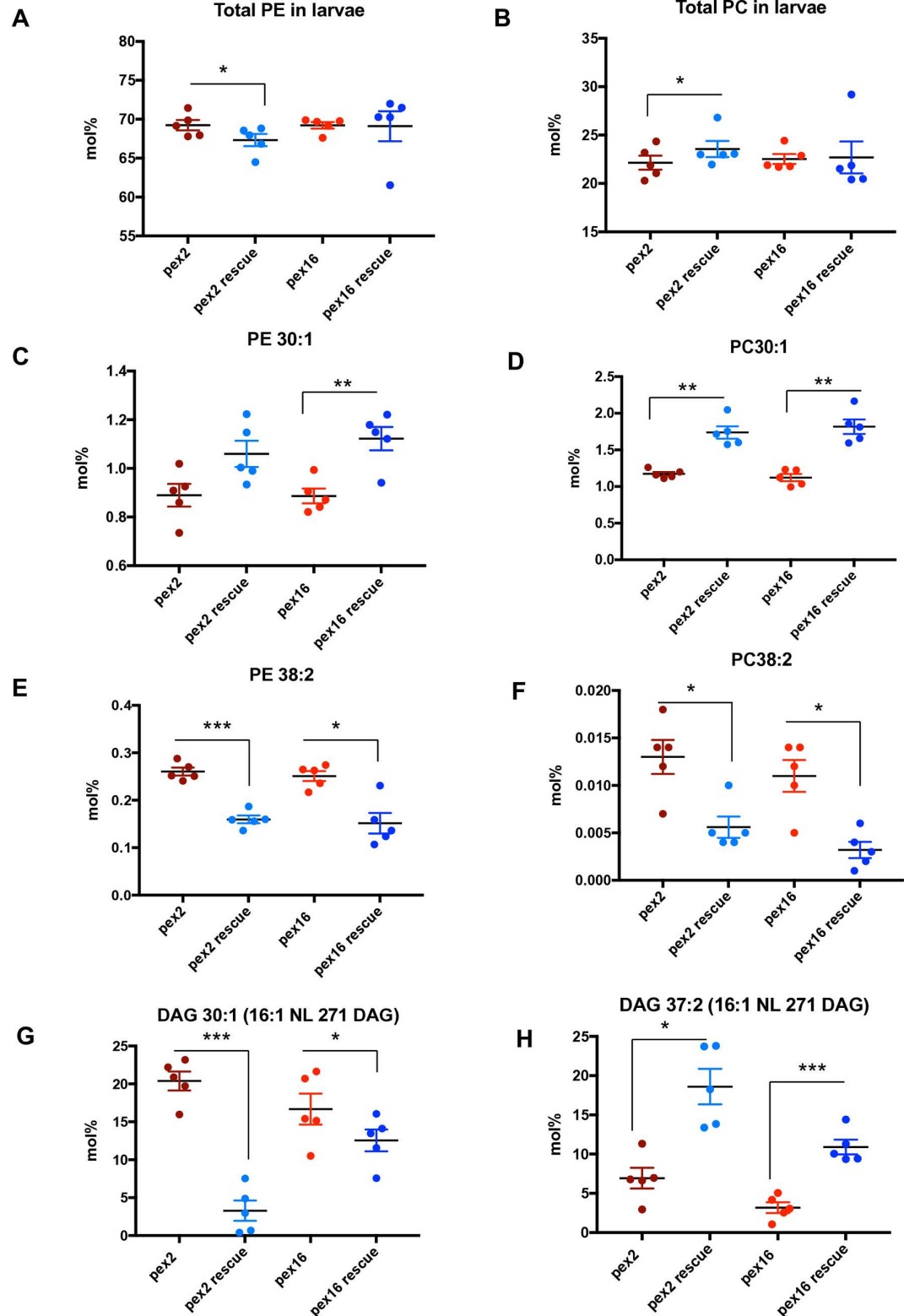

**Fig 2. Chain-length specific phospholipid and diacylglycerol abnormalities in peroxisomal mutant larvae.** A. Total levels in mol% of phosphatidylethanolamine (PE) in *pex2* and *pex16* larvae shows a mild but statistically significant increase in total PE in *pex2* mutant larvae compared to pex2 rescue (ratio *pex2/pex2 rescue*: 1.03, p = 0.0221), and no difference in PE in *pex16* mutant larvae compared to pex16 rescue (pex16/pex16 rescue: ratio

1.00, p = 0.94). B. Total levels in mol% of phosphatidylcholine (PC) in *pex2* and *pex16* larvae shows a mild but statistically significant decrease in total PC in *pex2* mutant larvae compared to *pex2* rescue (ration *pex2/pex2* rescue: ratio 0.94,p=0.038), and no difference in PC in *pex16* mutant larvae compared to *pex16* rescue (*pex16/pex16* rescue: ratio 0.99, p=0.897). C. Levels in mol% of PE 30:1 in *pex2* and *pex16* larvae shows no significant difference in PE 30:1 in *pex2* mutant larvae compared to pex2 rescue (ratio *pex2/pex2 rescue*: 0.839, p=0.068), and a significant reduction in *pex16* mutant larvae compared to pex16 rescue (pex16/pex16 rescue: ratio 0.790, p=0.008). D. Levels in mol% of PC 30:1 in *pex2* and *pex16* larvae shows a significant decrease in PC 30:1 in *pex2* mutant larvae compared to pex2 rescue (ratio *pex2/pex2 rescue*: 0.675, p=0.0005), and a significant reduction in in *pex16* mutant larvae compared to *pex16* rescue (pex16/pex16 rescue: ratio 0.618, p=0.002). E. Levels in mol% of PE 38:2 in *pex2* and *pex16* larvae shows a significant increase in PE 38:2 in *pex2* mutant larvae compared to pex2 rescue (ratio *pex2/pex2 rescue*: 1.63, p=0.0003), and a significant reduction in in *pex16* mutant larvae compared to *pex16* rescue (pex16/pex16 rescue: ratio 1.660, p=0.011). F. Levels in mol% of PC 38:2 in *pex2* and *pex16* larvae shows a significant increase in PC 38:2 in *pex2* mutant larvae compared to pex2 rescue (ratio *pex2/pex2 rescue*: 2.303, p=0.043), and a significant reduction in *pex16* mutant larvae compared to *pex16* rescue (pex16/pex16 rescue: ratio 3.24, p=0.019). G. Levels in mol% of Diacylglycerol 30:1 (16:1 NL 271 DAG) in *pex2* and *pex16* larvae shows a significant increase in DAG 30:1 in *pex2* mutant larvae compared to pex2 rescue (ratio *pex2/pex2 rescue*: 6.197, p=0.0006), and a significant reduction in in *pex16* mutant larvae compared to *pex16* rescue (pex16/pex16 rescue: ratio 1.328, p=0.0477). H. Levels in mol% of Diacylglycerol 37:2 (16:1 NL 271 DAG) in *pex2* and *pex16* larvae shows a significant decrease in DAG 37:2 in *pex2* mutant larvae compared to pex2 rescue (ratio *pex2/pex2 rescue*: 0.373, p = 0.0257), and a significant reduction in in *pex16* mutant larvae compared to *pex16* rescue (pex16/pex16 rescue: ratio 0.291, p=0.0003).

groups. However, we observed a pattern specific to 28 and 30 carbon that differed from 38 carbon, for example. In order to characterize this, we distinguish between broad classes of long chain phospholipids and so we term a phospholipid to have an intermediate chain length if the total carbon chain (across the two acyl chains) is C28-C31, and we consider C32 and above to be long chain length phospholipids. Indeed, a number of intermediate chain length PC and PE's exhibit the same pattern. PC 28:1 and PC 28:0 are both reduced in *pex2* mutants compared to rescue and *pex16* mutant compared to rescue (Fig S2A). PE 28:1 and PE 28:0 also show reduction in the *pex2* and *pex16* mutants compared to rescue larvae (Fig S2B). Reduced levels of other intermediate chain phospholipids such as PC 30:2, PC 30:0, PC 31:2, PC 31:1 and PC 31:0 are also seen (Fig S2C).

In contrast, long chain length PCs and PEs exhibit the opposite effect. Both *pex2* and *pex16* mutants have increased PE 38:2 and PC 38:2 phospholipids (**Fig 2E** **and** **2F**). Likewise PC 37:2 is also increased in peroxisome deficient larvae (Fig S3). These data suggest that while the overall pool of PCs and PEs in peroxisome deficient mutant larvae appears similar to controls and rescue, there is a deficiency of intermediate chain-length species and excess of long chain lengths. We therefore hypothesized that in peroxisomal mutants due to the excess of very long chain fatty acids, an over-abundance of long acyl chains would result.

One possibility to explain the increased abundance of long chains and reduced intermediate chain lengths is that altered chain lengths of the precursors of PCs and PEs might produce an imbalance in downstream products. We therefore examined diacylglycerols (DAGs) which can be converted to PCs and PEs through the Kennedy pathway [12]. Counter to our initial expectation, we observed an apparent excess of intermediate chain such as DAG 30:1 in both *pex2* and *pex16* mutants (**Fig 2G**). Long chain length DAGs such as DAG 37:2 were reduced in *pex2* and *pex16* mutants (**Fig 2H**). These data suggest that intermediate chain length phospholipid DAGs are actually increased in the presence of decreased PC and PEs in *pex* mutants. Thus, the DAG precursors exhibit an opposite pattern compared to PE and PC in terms to which chain lengths are most abundant in peroxisomal mutants.

The overall patterns are apparent when looking across multiple lipids, while individual lipids could vary. We plotted the ratio of specific phospholipids for *pex2* mutants compared to *pex2* rescue and *pex16* mutants compared to *pex16* rescue. The ratio of PCs across a range of chain lengths from PC 28:1 to PC 38:1 showed an increase in PC amounts as chain length increases (**Fig 3A**), with similar findings for PEs (**Fig 3B**). These ratios show that the phospholipids have an imbalance toward long chain lengths in the mutants. In contrast, DAG ratios in *pex* mutants have an imbalance toward intermediate chain lengths (**Fig 3C**). These observations have some similarities to the cellular data of patients with PBD previously reported [8], and in those studies, not only chain-length but also the number of unsaturations in the carbon chain have also been observed to play a role. We therefore examined the phospholipids according to how many

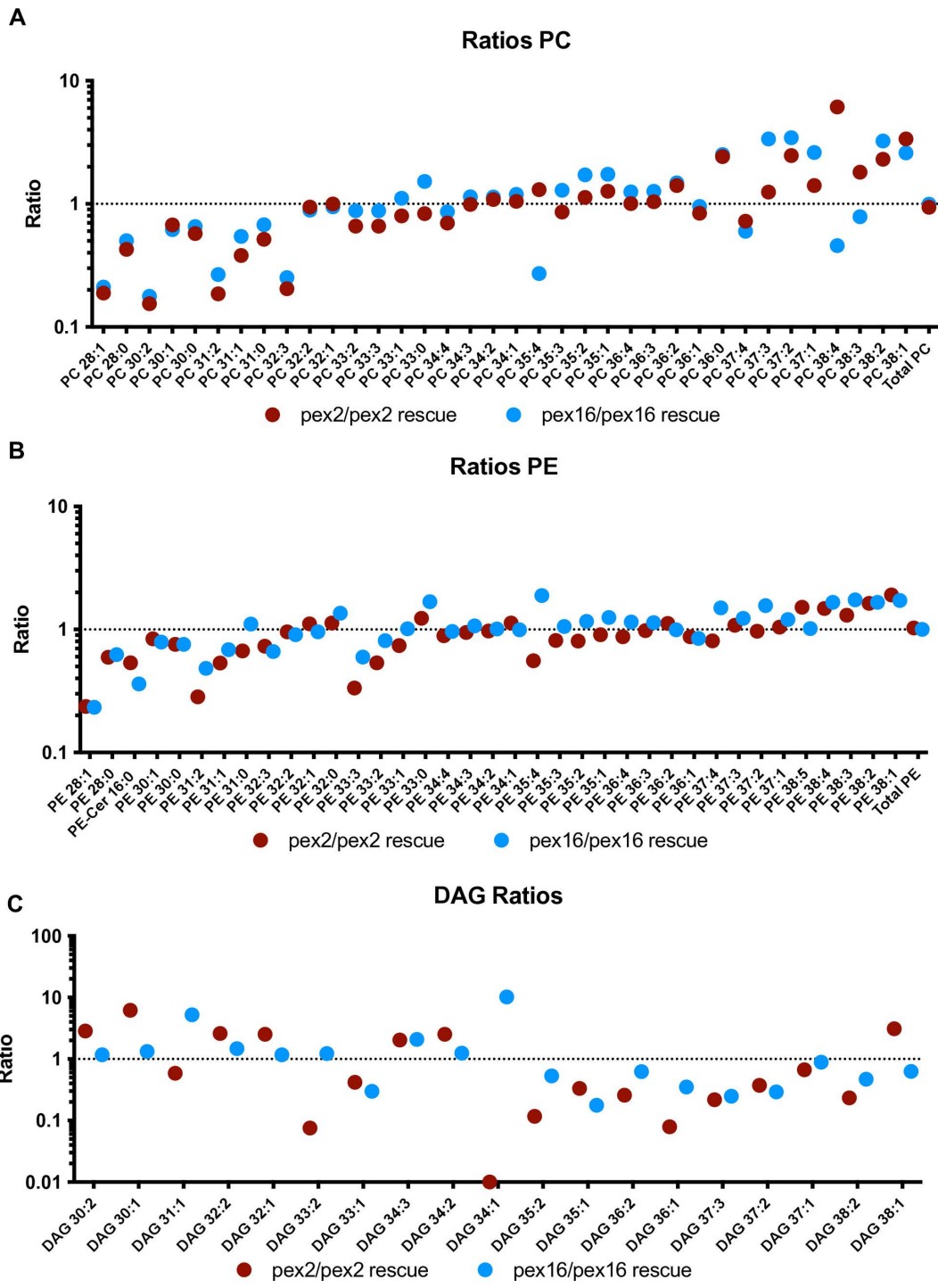

**Fig 3. Inverse relationship between phospholipid and diacylglycerol ratios spanning intermediate and long chain lengths.** A. Ratios of PCs comparing *pex2/pex2* rescue (red) and *pex16/pex16* rescue (yellow). These ratios are notably low for intermediate chain length PCs. For PC 28:1, pex2/pex2 rescue ratio 0.1889, pex16/pex16 rescue ratio 0.2111. For PC 30:2 *pex2/pex2* rescue ratio 0.154, *pex16/pex16* rescue ratio 0.177. These ratios gradually increase with variability as the chain length increases. For PC 38:2, pex2/pex2 rescue ratio 2.303, pex16/pex16 rescue ratio 3.243. For PC 38:1, pex2/pex2 rescue ratio 3.372, pex16/pex16 rescue ratio 2.598. B. Ratios of PEs comparing pex2/pex2 rescue (red) and pex16/pex16 rescue (yellow). These ratios are notably low for intermediate chain length PEs. For PE 28:1, pex2/pex2 rescue ratio 0.2369, pex16/pex16 rescue ratio 0.2328. For PE

31:2 *pex2/pex2* rescue ratio 0.2839, pex16/pex16 rescue ratio 0.4841. These ratios gradually increase with variability as the chain length increases. For PE 38:2, *pex2/pex2* rescue ratio 1.631, pex16/pex16 rescue ratio 1.660. For PE 38:1, pex2/pex2 rescue ratio 1.906, pex16/pex16 rescue 1.723.C. Ratios of DAGs comparing *pex2/pex2* rescue (red) and *pex16/pex16* rescue (yellow). These ratios are notably high for intermediate chain length (in contrast to PEs and PCs). For DAG 30:1 *pex2/pex2* rescue ratio 6.197, pex16/pex16 rescue ratio 1.328. DAG 32:2 *pex2/pex2* rescue ratio 2.601 pex16/pex16 rescue ratio 1.474. These ratios gradually decrease with variability as the chain length increases (in contrast to the increases seen in PEs and PCs). For DAG 37:2, *pex2/pex2* rescue ratio 0.3726, pex16/pex16 rescue ratio 0.2913. For DAG 38:2, pex2/pex2 rescue ratio 1.906, pex16/pex16 rescue 1.723.

unsaturations in *Drosophila* mutants (Fig S4). For example examining PC 30:0, PC 32:0 and PC 34:0 as a group of PCs with zero unsaturations (e.g., PC N:0). The ratio of *pex2* and *pex16* mutants compared to rescue show increases as chain length increases across different numbers of unsaturations such as PC N:0, PC N:1, PC N:2 and PC N:3 phospholipid (where N represents the carbon chain of different lengths), but the number of unsaturations themselves did not appear to have a dramatic impact on levels in *Drosophila pex* mutant larvae (Fig S4). In summary, the *Drosophila* larvae have PE and PC with longer carbon chain lengths and DAG precursors with more intermediate carbon chain lengths. Moreover, in *Drosophila* larvae the number of unsaturations does not appear to be dramatically altered in the mutants.

## Phospholipid alterations in the brain of Drosophila pex mutant adult flies

Next we examined the adult fly brain to assess the neuro-metabolic impact in *pex2* and *pex16* mutants (Table S3). In contrast to larvae where the relative amounts of PE and PC were similar between *pex* mutants and rescue lines, altered relative amounts of total PE and PC were observed in adult heads. For *pex2*, both mutant alleles were associated with reduced relative amounts of total PE compared to rescue lines, although the mutants' PE levels were not significantly different than the control line. This discrepancy could be due to the control line not matching the genetic background, and in general the genomic rescue line could be regarded as a more appropriate control for this background (Fig 4A). Similarly for *pex16*, reduced total PE is observed in the *pex16$^1$* allele compared to rescue, with a much less dramatic (though still significant) difference between the hypomorph *pex16$^{EY}$* line and rescue (Fig 4A). The difference between the two *pex16* mutant lines is likely due to comparison between a null and hypomorphic allele [13]. For total PC in the adult brain there were clear differences between all the pex2 alleles with higher levels of PC comopared to control and rescue lines (Fig 4B). For pex16 there was a complex range of results for the different genotypes (Fig 4B). Taken together these data suggest there may be an overall reduction in total PE in adult *Drosophila* brain, although variability amongst control and rescue lines lead to uncertainty about the biological significance of these alterations. The most consistent phospholipid alterations in *Drosophila* brain could be grouped according to PCs were consistently increased in peroxisomal mutants such as PC 30:0, PC 35:1, PC 34:1, PC 34:2, PC 37:2 and PC 37:1 (Fig 4C–4F, Fig S5). PE's exhibited different patterns with consistent decreases in PE 33:3, PE 33:2, PE 33:1, PE 35:4, PE 35:3 and PE 35:2 (Fig S6). Consistent increases in PE were observed in PE 33:0, PE 34:4, PE 34:3, PE 34:1, PE 36:3 and PE 38:3 (Fig S7). These findings pointed to phospholipid chains where the number of unsaturations could influence whether the levels were increased or decreased. Overall, these findings suggest altered chain length of PEs and PC's in adult fly brain.

## Altered PE ceramides in pex mutant brains

In a previous metabolomic analyses in patients with PBD-ZSD, typically with mutations in the *PEX1* gene, we observed reduced sphingomyelin levels [9]. Since we observed reduced levels of intermediate chain length phospholipids in *Drosophila* in this study, we considered whether a related abnormality could underlie the sphingolipid reductions we observed in the human studies. Sphingomyelins are derived from phosphatidylcholine. However, the acyl chain in sphingomyelins are not derived from PC. Sphingomyelin receives the phosphocholine headgroup from PC and the carbon chains in sphingomyelin are derived from ceramide. Therefore, the precursor determining acyl chain length differs between sphingomyelin and phospholipids. Moreover, insects do not have prominent levels or biological roles for sphingomyelin

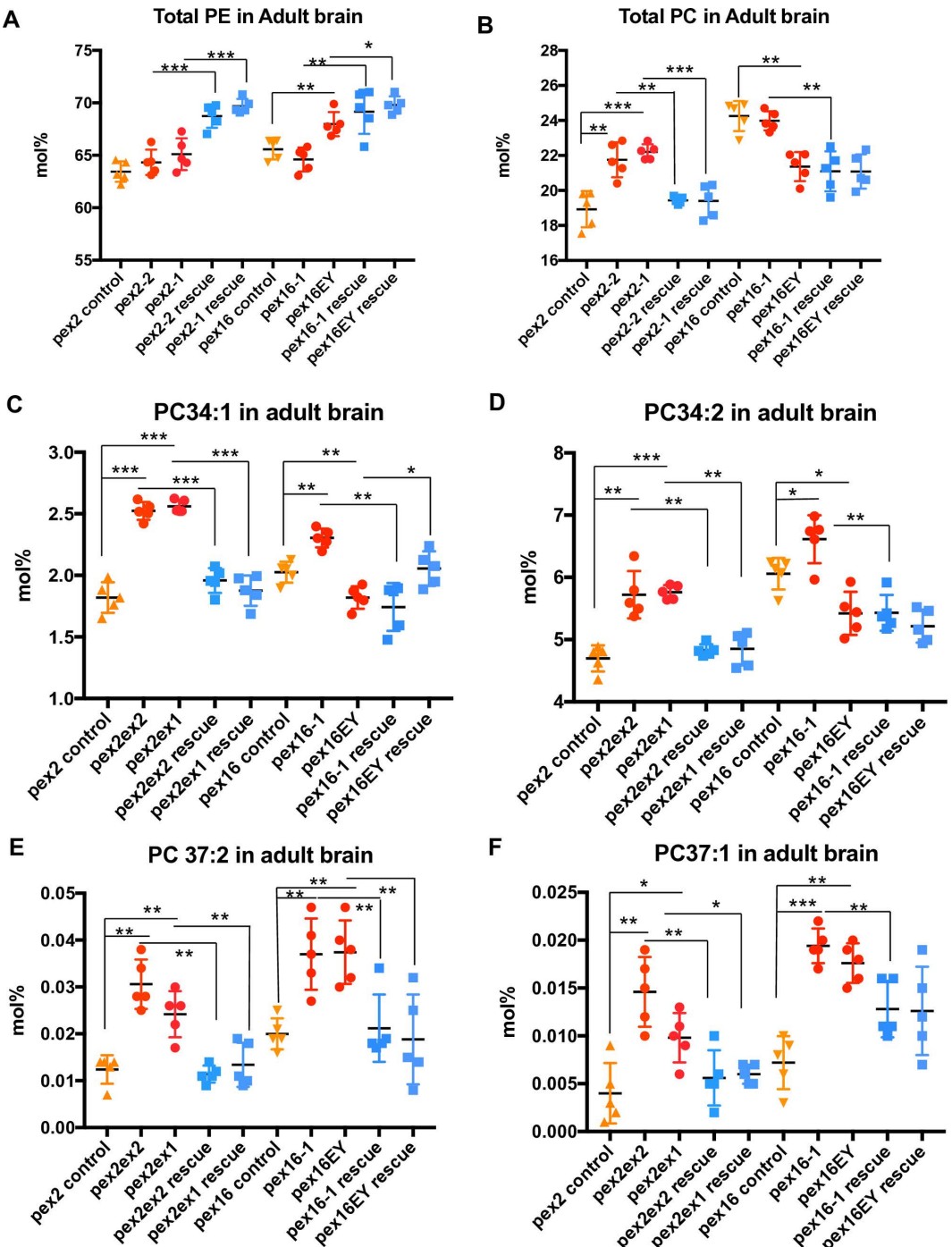

**Fig 4. Chain-length specific phospholipid and diacylglycerol abnormalities in peroxisomal mutant brain.** A. Total levels in mol% of Phosphati-dylethanolamine (PE) in *pex2* and *pex16* adult brains (genotypes shown in Fig 1) shows a reduction in total PE in *pex2* mutant brain compared to pex2 rescue (ratio *pex2²/pex2²* rescue 0.936, p=<0.0001; ratio *pex2¹/pex2¹* rescue 0.934, p=0.001), and reduced PE in *pex16* mutant larvae compared to pex16 rescue (*pex16¹/pex16¹* rescue 0.934, p=0.005; ratio pex16EY/Pex16EY rescue 0.974, p=0.0226). B. Total levels in mol% of Phosphatidylcho-line (PC) in *pex2* and *pex16* adult brains (genotypes shown in Fig 1) shows increased levels in total PC in pex2 mutant brain compared to pex2 rescue (ratio *pex2²/pex2²* rescue 1.119, p=0.006; ratio *pex2¹/pex2¹* rescue 1.144, p=0.001), but the effect on *pex16* mutant larvae is complex with the significant differences between *pex16^{EY}* as being reduced compared to control (but not *pex16^{EY}* rescue) and *pex16¹* levels being increased compared to *pex16¹* rescue (but not control). C. Levels in mol% of PC 34:1 in *pex2* and *pex16* mutant brains shows significant increases in *pex* mutant brains. For PC 34:1

in *pex2²* brain and *pex2¹* brain, levels are increased compared to controls (ratio 1.387, 1.407 respectively, p<0.001, p<0.001 respectively) and *pex2²* brain compared to rescue (ratio 1.289, p<0.001) as well as *pex2¹* compared to rescue (ratio 1.365, p<0.001). For PC 34:1 in *pex16¹* levels are increased compared to controls (ratio 1.137, respectively, p<0.001, p<0.001 respectively) and rescue (ratio 1.321, p=0.002). However for *pex16^{EY}* brain the levels are mildly reduced compared to control (ratio 0.898, p=0.006), and *pex16^{EY}* compared to rescue (ratio 0.885, p=0.017). D. Levels in mol% of PC 34:2 in *pex2* and *pex16* mutant brains shows significant increases in *pex* mutant brains. For PC 34:2 in *pex2²* brain and *pex2¹* brain, levels are increased compared to controls (ratio 1.217, 1.226 respectively, p=0.002, p<0.001 respectively) and *pex2²* brain compared to rescue (ratio 1.184, p=0.005) as well as *pex2¹* compared to rescue (ratio 1.188, p=0.001). For PC 34:2 in *pex16¹* levels are increased compared to controls (ratio 1.092, p=0.031) and compared to rescue (ratio 1.218, p=0.001). However, *pex16^{EY}* brain has reduced levels compared to controls (ratio 0.895, p=0.012). and no difference from rescue. E. Levels in mol% of PC 37:2 in *pex2* and *pex16* mutant brains shows significant increases in *pex* mutant brains. For PC 37:2 in *pex2²* brain and *pex2¹* brain, levels are increased compared to controls (ratio 2.471, 1.939 respectively, p=0.001, p=0.004 respectively) and *pex2²* brain compared to rescue (ratio 2.722, p=0.001) as well as *pex2¹* compared to rescue (ratio 1.806, p=0.008). For PC 37:2 in *pex16¹* and *pex16^{EY}* brain levels are increased compared to controls (ratio 1.860, 1.871 respectively, p=0.005, p=0.003 respectively) and *pex16¹* compared to rescue (ratio 1.755, p=0.009) and for *pex16^{EY}* compared to rescue (ratio 2.001, p=0.009). F. Levels in mol% of PC 37:1 in *pex2* and *pex16* mutant brains shows significant increases in *pex* mutant brains. For PC 37:1 in *pex2²* brain and *pex2¹* brain, levels are increased compared to controls (ratio 3.770, 2.504 respectively, p=0.001, p=0.010 respectively) and *pex2²* brain compared to rescue (ratio 2.621, p=0.003) as well as *pex2¹* compared to rescue (ratio 1.628, p=0.022). For PC 37:1 in *pex16¹* and *pex16^{EY}* brain levels are increased compared to controls (ratio 2.703, 2.442 respectively, p<0.001, p<0.001 respectively) and *pex16¹* compared to rescue (ratio 1.519, p=0.003) and non-significant increase for *pex16^{EY}* compared to rescue (ratio 1.398, p=0.065).

(PC-ceramides). In insects such as *Drosophila*, PE-ceramides are of greater importance and abundance in insect membranes and, instead of an 18C-sphingoid base, contain a 14C sphingoid base [23]. We therefore examined PE-ceramides in the fly brains. We observed that PE Cer 22:0 is reduced in *pex2* mutants (**Fig 5A**) and in *pex16* mutants (**Fig 5B**), although the hypomorphic *pex16^{EY}* allele does not show this effect. In contrast, longer chain length PE Cer 26:0 is increased in *pex2* mutants (**Fig 5C**) as well as *pex16* mutants (**Fig 5D**). In summary, we observe altered chain length of PE ceramides in the adult fly brain. This alteration is an imbalance toward longer chain lengths in fly brains.

### Carbon chain length affects the levels of sphingomyelin in human subjects

Having observed a pattern of altered carbon chain lengths in *pex* mutant *Drosophila* we sought to confirm these observations in human samples. We examined the levels of PC, and sphingomyelin in human plasma samples (Table S4). This included 16 individuals with PBD-ZSD due to *PEX1* genetic changes, the clinical details of these subjects has been previously reported [9]. We compared these samples to pediatric disease samples without peroxisomal disease from a distinct genetic cohort with no known peroxisomal link [24], which we termed pediatric controls and we also tested unaffected adult controls. In plasma we observed increased relative levels of PC (Fig 6A). We also observed a decrease in the total levels of plasma ether-linked PC, i.e., alkyl or plasmenyl PC, here referred to as "ePC" (Fig 6B), a finding likely related to plasmalogen defects in PBD. Chain-length abnormalities were also observed, including decreased levels of PC 34:2 (Fig 6C), PC(36:2) (Fig 6D), but an increase in PC(38:2) (Fig 6E), and PC(38:1) (Fig 6F) in patients with *PEX1* mutations compared to controls. Increased levels of lysophosphatidylcholine were also observed in plasma (Fig S8). Indeed, a modest effect on PC molecular species according to chain length was observed in the human plasma samples with the PBD-ZSD patients exhibiting high levels of PC 40:3, PC 38:1, PC 38:2 and the low levels of PC 34:0 and PC 36:4.

In sphingomyelin, we observed a reduction in SM 18:1 and SM 18:0 (**Fig 7A and 7B**) in patients with *PEX1* mutations. SM 22:0 was also reduced in the patients with PEX1 mutations (**Fig 7C**). However, for SM 24:0 the levels in plasma were the same for controls and the PBD-ZSD subjects (**Fig 7D**). In summary, there is a reduced level of certain subtypes of sphingomyelin in plasma with normal levels of very long chain sphingomyelins.

### Discussion

In this study we present a series of detailed lipidomic analyses on *Drosophila pex* mutants and human plasma samples from patients with PBD-ZSD. The principal findings of our study are that peroxisomal biogenesis mutations lead to reproducible alterations in phospholipids and sphingolipids in *Drosophila* larvae, brain and human plasma. These alterations

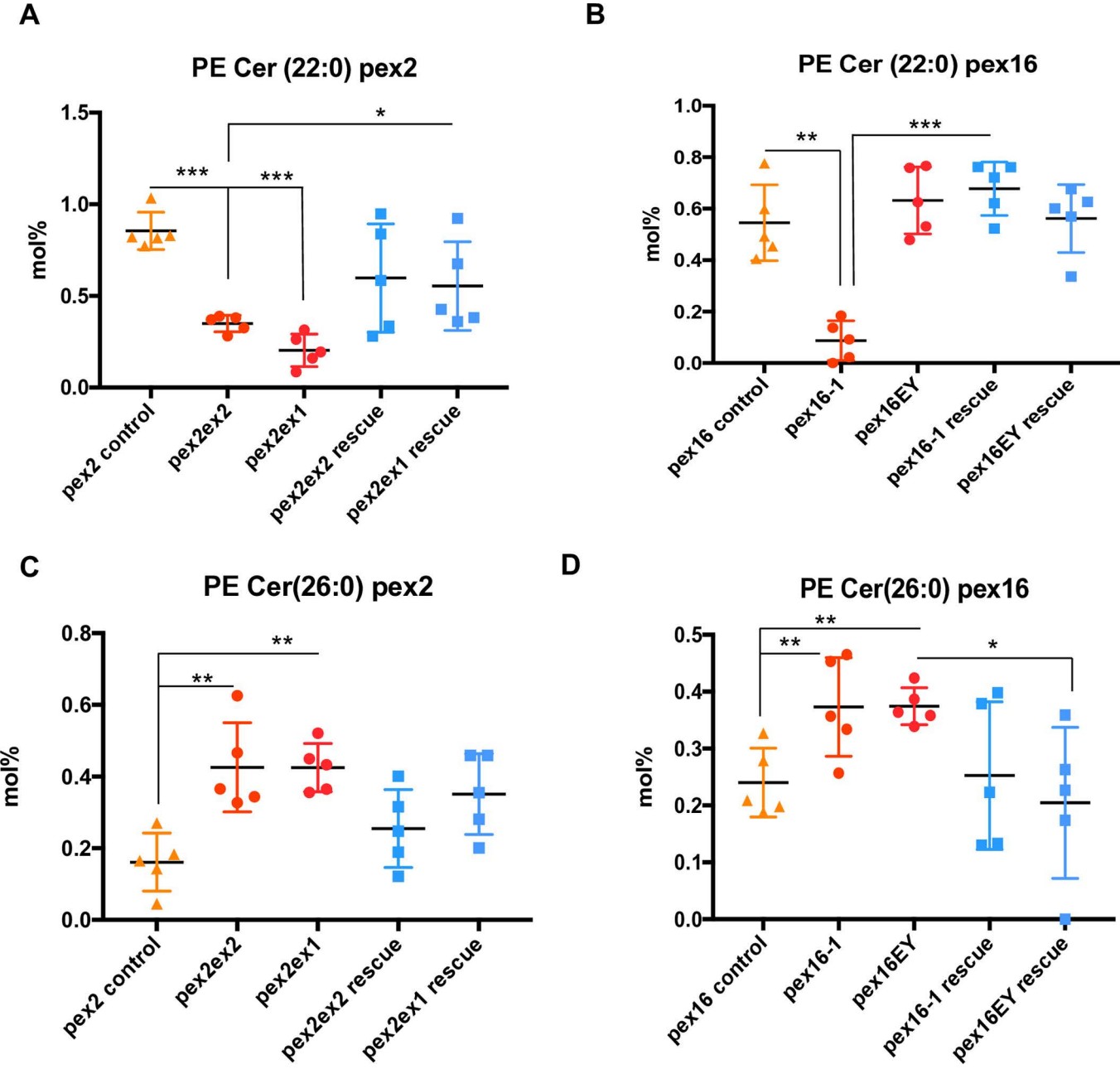

**Fig 5. Chain-length specific PE ceramide abnormalities in peroxisomal mutant brain. A.** Total levels in mol% of PE Cer 22:0 in *pex2* adult brains (genotypes shown in Fig 1) shows reduced levels in total PE Cer 22:0 in pex2 mutant brain compared to control (ratio *pex2²ⁱ/control* 0.409, ratio *pex2¹ⁱ/control* 0.238, p < 0.001, p < 0.001 respectively), and shows reduced levels in total PE Cer 22:0 compared to pex2 rescue for *pex 2¹* allele (ratio *pex-2¹ⁱ/pex2¹* rescue 0.368, p = 0.028). **B.** Total levels in mol% of PE Cer 22:0 in *pex16* adult brains (genotypes shown in Fig 1) shows reduced levels in total PE Cer 22:0 in *pex16¹* mutant brain compared to control (ratio *pex16¹ⁱ/control* 0.160, p=0.001), and *pex16¹* mutant brain compared to rescue (ratio 0.129, p<0.001) but no significant differences were observed for *pex 16ᴱʸ* allele. **C.** Total levels in mol% of PE Cer 26:0 in *pex2* adult brains (genotypes shown in Fig 1) shows increased levels in total PE Cer 26:0 in pex2 mutant brain compared to control (ratio *pex2²ⁱ/control* 2.643, p=0.005, ratio *pex2¹ⁱ/control* 2.638, p<0.001, p=0.001 respectively), and increased on average compared to rescue although the differences were not significant. **D.** Total levels in mol% of PE Cer 26:0 in *pex16* adult brains (genotypes shown in Fig 1) shows increased levels in total PE Cer 26:0 in pex16 mutant brain compared to control (ratio *pex16¹ⁱ/control* 1.552, p=0.026, and increased fro *pex16ᴱʸ* compared to control and to rescue (ratio *pex16ᴱʸ*/control ratio 1.557, p=0.004, ratio *pex16ᴱʸ*/rescue ratio 1.831, p=0.0439).

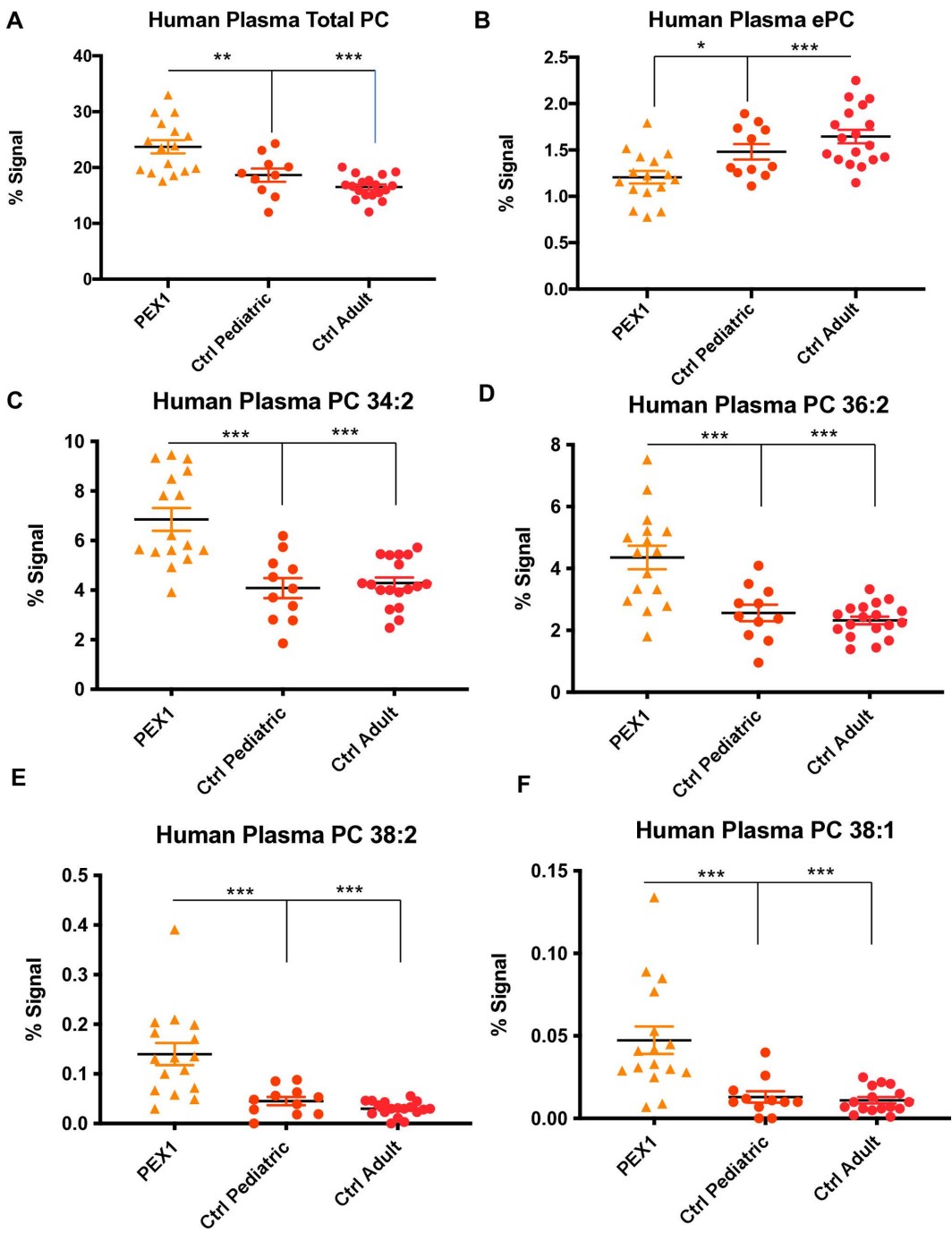

**Fig 6. Phospholipid abnormalities in human plasma from patients with PBD-ZSD. A.** Human plasma PC (as %) is increased in patients with PEX1 mutations compared to pediatric (ratio 1.26, p = 0.005) and adult controls (ratio 1.437, p < 0.001). **B.** Human plasma ether linked phosphatidylcholine (ePC) is reduced in patients with PEX1 mutations compared to pediatric (ratio 0.815, p=0.018) and adult controls (ratio 0.733, p<0.001). **C.** Human plasma PC 34:2 is increased in patients with PEX1 mutations compared to pediatric (ratio 1.678, p<0.001) and adult (ratio 1.600, p<0.001) controls. **D.** Human plasma PC 36:2 is increased in patients with PEX1 mutations compared to pediatric (ratio 1.702, p<0.001) and adult controls (ratio 1.878, p<0.001). **E.** Human plasma PC 38:2 is increased in patients with PEX1 mutations compared to pediatric (ratio 3.109, p<0.001) and adult controls (ratio 4.729, p<0.001). **F.** Human plasma PC 38:1 is increased in patients with PEX1 mutations compared to pediatric (ratio 3.679, p = 0.001) and adult controls (ratio 3.784, p < 0.001).

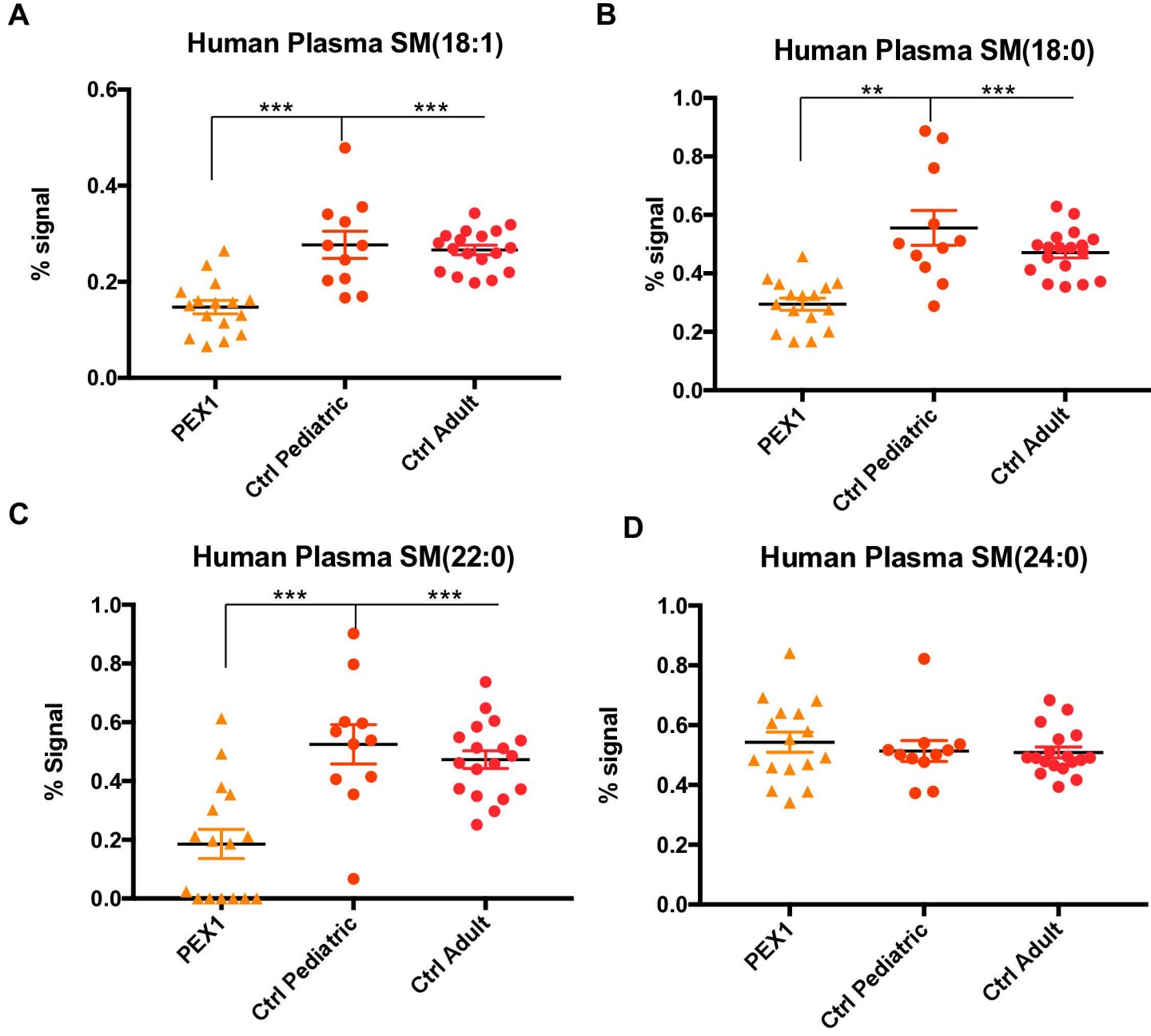

**Fig 7. Sphingolipid abnormalities in human plasma from patients with PBD-ZSD. A.** Human plasma SM 18:1 is decreased in patients with PEX1 mutations compared to pediatric (ratio 0.532, p < 0.001) and adult controls (ratio 0.553, p < 0.001). **B.** Human plasma SM 18:0 is decreased in patients with PEX1 mutations compared to pediatric (ratio 0.530, p=0.001) and adult controls (ratio 0.626, p<0.001). **C.** Human plasma SM 22:0 is decreased in patients with PEX1 mutations compared to pediatric (ratio 0.354, p=0.001) and adult controls (ratio 0.393, p<0.001). **D.** Human plasma SM 24:0 does not have differences for patients with PEX1 mutations compared to pediatric and adult controls.

have some consistent features in all these tissues including an overabundance of longer chain phospholipids like PC and PE and reduced relative amount of intermediate chain lengths. This unique *Drosophila* dataset is complementary with the findings related to human cells and tissues [8,11]. However, it also points to differences in the relative distribution of specific carbon chain-lengths, for example the overabundance of intermediate acyl chains in DAGs and their relative

reduction in PEs and PCs. As DAGs are part of the Kennedy pathway for de novo synthesis of PEs and PCs, the imbalanced distribution suggests that regulation or effects such as substrate shuttling rather than simple chemical equilibrium are important factors to consider in PBD.

Defects in peroxisomal biogenesis have a well-documented overabundance of very long acyl chains as peroxisomes are required for the catabolism of very long chain fatty acids. In our dataset we observe increased long chain PCs and PEs but reduced long chain DAGs. The data are analyzed in mol% and therefore the overabundance of one analyte will lead to reduction in the percent of other analytes and so the increased long chain PCs and PEs are accompanied by reduced intermediate chain. However, for DAGs the inverse pattern, increased intermediate chains and reduced long chains are observed. PCs and PEs can be derived from diacyglycerol through the Kennedy pathway a process of *de novo* biosynthesis of phospholipids [12]. DAGs are thus converted to phospholipids via the Kennedy pathway. If these two pools of lipids (PEs and PCs versus DAGs) were in simple chemical equilibrium, the excess intermediate chain DAGs could be converted to intermediate chain PC and PEs and the long chain phospholipids could be converted to long chain DAGs and the imbalance we observe in the *pex* mutants would not exist. The fact that these imbalances are observed suggests that these lipid pools are not in equilibrium and this is an effect of substrate shuttling, a phenomenon in which certain substrates are preferentially incorporated into products in fatty acid pathways [25]. The Kennedy pathway is not the only determinant of acyl chain length in phospholipids as specific acyl chains can be incorporated into PC and PE through exchange and remodeling via the Land's cycle [26,27]. The Kennedy pathway for de novo biosynthesis through DAGs has preferential incorporation of the intermediate chain acyl chains into the resulting PEs and PCs due to preference of glycerol-3-phosphate acyltransferases (GPAT) [28]. However, the remodeling of the phospholipid through the Land's cycle which exchanges acyl chains on the glycerol backbone has a preference for very long and long chains of some of the acyl transferases [27].

We propose a model for phospholipid and sphingolipid abnormalities in PBD (Fig 8). We base this model on the following observations: 1) peroxisomes are required for very long chain fatty acid breakdown and peroxisome dysfunction will lead to a long-term excess of very-long acyl chains 2) the excess of very-long acyl chains will be incorporated into many classes of phospholipids and sphingolipids and indeed increased levels of C24 and C26 derivatives of many biochemical lipids have been previously observed and 3) phospholipids in particular have two pathways, the Kennedy pathway which has a preference for generating PCs and PEs with intermediate acyl chains, and the Land's cycle which has preference for very long and long chains. In our model the normal physiologic balance of acyl chains within PE, PC and DAGs and thus the mol% values in normal cells results from an overall balance in the acyl chain pool with some shuttling in Kennedy pathway of intermediate chains and some shuttling of very long and long chains in the Land's cycle (Fig 8A). In PBD in contrast, the overall imbalance of acyl chains due to the peroxisomal fatty acid oxidation defect produces downstream phospholipid abnormalities that we observe across phospholipids and sphingollipids (Fig 8B). The data we present here has some limitations in that the biochemical measurements at one point in time cannot provide information on the kinetics of enzymatic reactions in a pathway. However, our model explains a number of observations we have previously been unable to account for. It explains why we had previously observed normal overall levels of PC and PE but unexplained increases in phosphocholine and phosphoethanolamine [13] (Fig S1) as our model predicts that reduced flux through the Kennedy pathway would result. In addition, this data suggests an explanation for our report that sphingomyelin levels are reduced in plasma in PBD-ZSD [9]. Our data on PE-ceramides in *Drosophila* models show a similar altered distribution of acyl chain length similar to what we see in PE and PC (Fig 8). In *Drosophila* and other insects PE ceramides are the predominant sphingolipid rather than sphingomyelins [29]. In these lipids the carbon chain lengths are derived from ceramides. There is likely a very similar abnormality in sphingomyelins in plasma (Fig 7). Similar to phospholipids the excess levels of very long chain lipids in peroxisomal mutants lead to overabundance of very long chain sphingolipids such as sphingomyelin and PE ceramides and reduced overall levels of intermediate chain. This therefore explains the apparent reduced levels in the intermediate chains we previously reported [9]. Indeed, it was the *Drosophila* model allowed us to

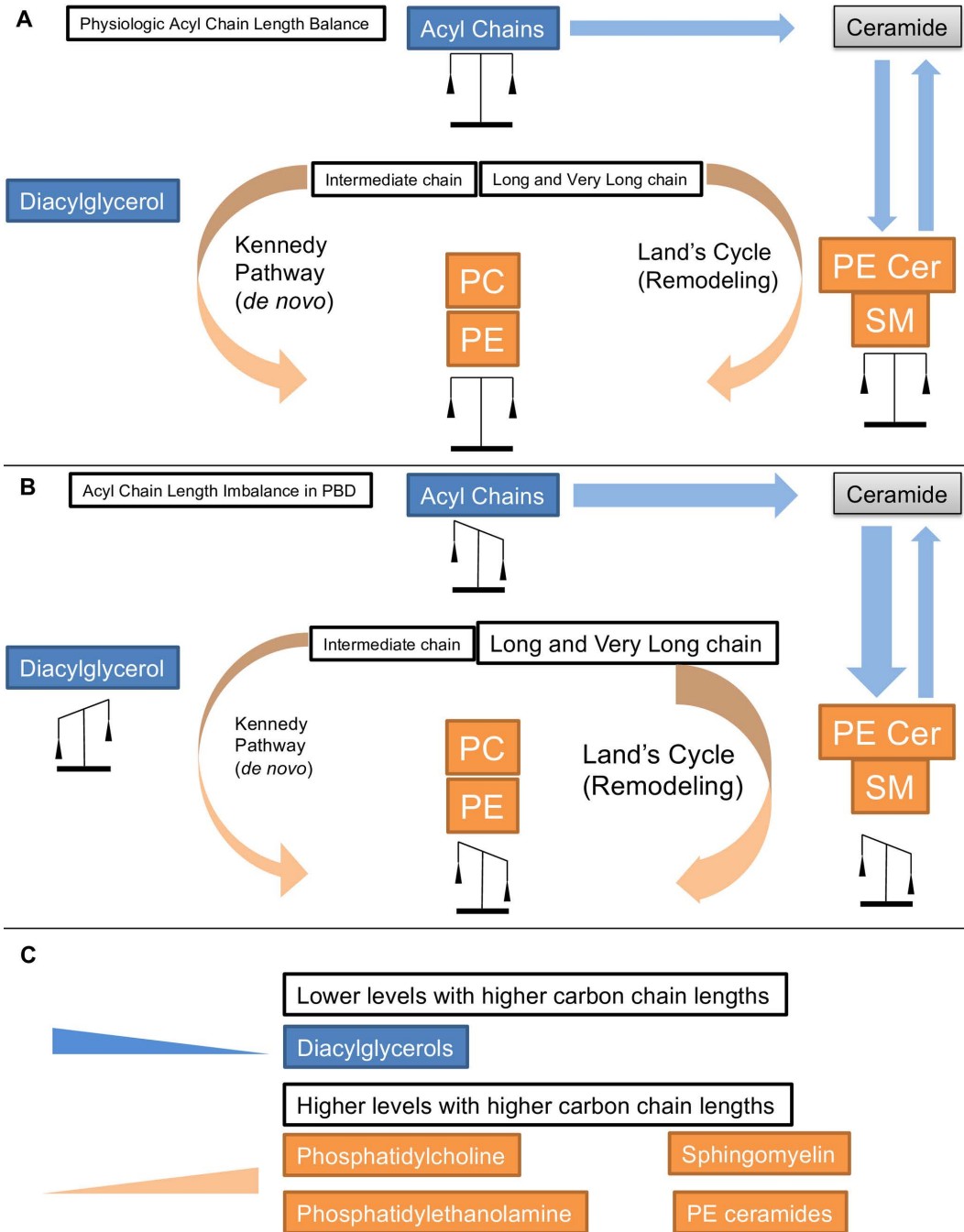

**Fig 8. Biochemical model of phospholipid and sphingolipid abnormalities for peroxisomal biogenesis disorders.** The Kennedy pathway mediates generation of phosphatidylcholine and phosphatidylethanolamine from diacylglycerol. Peroxisomal biogenesis defects lead to an excess of long and very long chain PC and PE, but an apparent reduction in intermediate chain. However the diacylglycerol levels appear to have an inverse relationship with reductions of very long and long chains and increased levels of intermediate chains. For sphingolipids (where the chain lengths are notably not in direct equilibrium with phospholipids) a similar excess of long and very long chains and reduced levels of intermediate chain. This loss of intermediate chains may explain the overall reduced levels of sphingomyelin observed in clinical samples.

observe a dramatic overabundance of long chain PE ceramides and reduced levels of intermediate chains and to infer the cause of the clinical observations in humans. This is supported by our data, although we cannot rule out an overall deficiency of sphingomyelins with a disproportionate quantity of the remaining having long or very long chain acyl chains. Our analysis therefore points to significant alterations in the most common membrane lipids which we see in the brain. The defect in peroxisomal biogenesis that underlies PBD-ZSD likely has a dramatic impact on the composition of neuronal membranes, features that should be further explored for potential clinical consequences in PBD-ZSD.

This study provides a large dataset of phospholipid measurements in *Drosophila pex* mutants with some comparative studies in human plasma samples. Lipidomic analyses measuring phospholipids have been done in human cell lines and samples and there are areas of overlap within our study on human plasma samples that suggest the *Drosophila pex* mutants model many aspects of phospholipid biochemistry in PBD-ZSD. The *Drosophila* model allows us to systematically study the brain in adult flies and to provide observations from the brain *in vivo*.

## Supporting information

**Fig S1. Proposed phospholipid metabolomic alterations in *pex2* and *pex16 Drosophila* mutants from prior study.** In our previous metabolomic analysis of pex2 and pex16 mutants phospholipid abnormalities were identified which included a decrease in glycerol and glycerol-3-phosphate. Diacylglycerol is converted to phosphatidlycholine (with the addition of choline) or phosphatidylethanolamine (with the addition of ethanolamine) and the choline and ethanolamine precursors appear to be increased in the pex mutants. Also noted was an apparent excess of lysolipids and reduces levels of phospholipid breakdown products glycerophophorylcholine/glycerolethanolamine and glycerol 3-phosphate. Of note the levels of phosphatidylcholine and phosphatidylethanolamine themselves were normal, though the specific chain lengths were not analyzed in these previous studies.
(TIF)

**Fig S2. Robust chain-length specific reductions in intermediate chain phospholipids in peroxisomal mutant larvae.** A. Total levels in mol% of PC 28:1 and PC 28:0 Phosphatidylcholine (PC) in *pex2* and *pex16* larvae shows dramatic and significant decreases in intermediate chain phospholipids. For PC 28:1, *pex2* mutant larvae compared to *pex2* rescue (ratio *pex2*/*pex2* rescue: ratio 0.1889,p = 0.0006), and a dramatic decrease in *pex16* mutant larvae compared to *pex16* rescue (*pex16*/*pex16* rescue: ratio 0.211, p = 0.003). For PC 28:0, *pex2* mutant larvae compared to *pex2* rescue (ratio *pex2*/*pex2* rescue: ratio 0.427,p = 0.0002), and a non-significant decrease in *pex16* mutant larvae compared to *pex16* rescue (*pex16*/*pex16* rescue: ratio 0.502, p = 0.0555). B. Total levels in mol% of PE 28:1 and PE 28:0 Phosphatidylethanolamine (PE) in *pex2* and *pex16* larvae shows dramatic and significant decreases in intermediate chain phospholipids. For PE 28:1, *pex2* mutant larvae compared to *pex2* rescue (ratio *pex2*/*pex2* rescue: ratio 0.2369,p=0.0003), and a dramatic decrease in *pex16* mutant larvae compared to *pex16* rescue (*pex16*/*pex16* rescue: ratio 0.233, p=7.97E-05). For PE 28:0, *pex2* mutant larvae compared to *pex2* rescue (ratio *pex2*/*pex2* rescue: ratio 0.594,p=0.0027), and a non-significant decrease in *pex16* mutant larvae compared to *pex16* rescue (*pex16*/*pex16* rescue: ratio 0.624, p=0.0022). C. Total levels in mol% of PC 30:2, PC 30:0, PC 31:2, PC31:1, and PC 31:0 Phosphatidylcholine (PC) in *pex2* and *pex16* larvae showing dramatic and significant decreases in intermediate chain phospholipids. For PC 30:2, *pex2* mutant larvae compared to *pex2* rescue (ratio *pex2*/*pex2* rescue: ratio 0.1545,p = 0.0002), and a dramatic decrease in *pex16* mutant larvae compared to *pex16* rescue (*pex16*/*pex16* rescue: ratio 0.1775, p = 0.0018). For PC 30:0, *pex2* mutant larvae compared to *pex2* rescue (ratio *pex2*/*pex2* rescue: ratio 0.5748,p = 0.0303), and a dramatic decrease in *pex16* mutant larvae compared to *pex16* rescue (*pex16*/*pex16* rescue: ratio 0.6542, p = 0.3048). For PC 31:2, *pex2* mutant larvae compared to *pex2* rescue (ratio *pex2*/*pex2* rescue: ratio 0.1857,p = 0.0020), and a dramatic decrease in *pex16* mutant larvae compared to *pex16* rescue (*pex16*/*pex16* rescue: ratio 0.2664, p = 0.0427). For PC 31:1, *pex2* mutant larvae compared to *pex2* rescue (ratio *pex2*/*pex2* rescue: ratio 0.3812,p = 0.0003), and a dramatic decrease in *pex16* mutant larvae compared to *pex16* rescue

(*pex16*/*pex16* rescue: ratio 0.5449, p=0.0150). For PC 31:0, *pex2* mutant larvae compared to *pex2* rescue (ratio *pex2*/ *pex2* rescue: ratio 0.5149,p=0.0002), and a non-significant decrease in *pex16* mutant larvae compared to *pex16* rescue (*pex16*/*pex16* rescue: ratio 0.6767, p=0.2111).
(TIF)

**Fig S3. Robust chain-length specific elevations in long chain phospholipids in peroxisomal mutant larvae.** Total levels in mol% of PC 37:2: and PC 38:2 Phosphatidylcholine (PC) in *pex2* and *pex16* larvae shows dramatic and significant increases in long chain phospholipids. For PC 37:2, *pex2* mutant larvae compared to *pex2* rescue (ratio *pex2*/*pex2* rescue: ratio 2.473,p=0.0183), and a dramatic increase in *pex16* mutant larvae compared to *pex16* rescue (*pex16*/*pex16* rescue: ratio 3.446, p=0.002). For PC 38:2, *pex2* mutant larvae compared to *pex2* rescue (ratio *pex2*/*pex2* rescue: ratio 2.303,p=0.0426), and a non-significant decrease in *pex16* mutant larvae compared to *pex16* rescue (*pex16*/*pex16* rescue: ratio 3.243, p=0.0193).
(TIF)

**Fig S4. Increasing levels of PCs based on chain length across different classes of lipid side chains organized by unsaturations.** The same ratios depicted in Fig 3A is reorganized based on the total number of unsaturations, organized from (N:0) up to (N:4) showing that across each class the ratios or pex2/pex2 rescue and pex16/pex16 rescue increases as chain-length increases. Clear trends are not observed for the number of unsaturations themselves.
(TIF)

**Fig S5. Robust chain-length specific elevations in some intermediate chain and long chain phospholipids in peroxisomal mutant brain. A.** Levels in mol% of PC 30:0 in *pex2* and *pex16* mutant brains show significant increases in *pex* mutant brains. For PC 30:0 in *pex2²* brain and *pex2¹* brain, levels are increased compared to controls (ratio 1.131, 1.238 respectively, p=0.038, p=0.001 respectively) and *pex2²* brain compared to rescue (ratio 1.383, p<0.001), and *pex2¹* compared to rescue (ratio 1.483, p<0.001). For PC 30:0 in *pex16¹* and *pex16^{EY}* brain, levels are increased compared to controls (ratio 1.368, 1.107 respectively, p<0.001, p=0.001 respectively). And *pex16¹* brain compared to rescue (ratio 1.186, p=0.002), and increased *pex16^{EY}* compared to rescue (ratio 1.222, p=0.008). **B.** Levels in mol% of PC 35:1 in *pex2* and *pex16* mutant brains shows significant increases in *pex* mutant brains. For PC 35:1 in *pex2²* brain and *pex2¹* brain, levels are increased compared to controls (ratio 2.529, 1.653 respectively, p=0.003, p=0.004 respectively) and *pex2²* brain compared to rescue (ratio 1.985, p=0.005) as well as *pex2¹* compared to rescue (ratio 2.059, p=0.006). For PC 35:1 in *pex16¹* and *pex16^{EY}* brain, levels are increased compared to controls (ratio 2.129, 2.310 respectively, p<0.001, p<0.001 respectively). And *pex16¹* brain compared to rescue (ratio 1.629, p=0.019), and for *pex16^{EY}* compared to rescue (ratio 1.690, p<0.001).
(TIF)

**Fig S6. Robust chain-length specific reductions in some intermediate chain and long chain PE phospholipids in peroxisomal mutant brain. A.** Levels in mol% of PE 33:3 in *pex2* and *pex16* mutant brains show significant decreases in *pex* mutant brains. For PE 33:3 in *pex2²* brain and *pex2¹* brain, levels are dramatically reduced compared to controls (ratio 0.225, 0.200 respectively, p<0.001, p<0.001 respectively) and *pex2²* brain compared to rescue (ratio 0.244, p<0.001), and *pex2¹* compared to rescue (ratio 0.220, p<0.001). For PE 33:3 in *pex16¹* and *pex16^{EY}* brain, levels are decreased compared to controls (ratio 0.214, 0.718 respectively, p<0.001, p<0.001 respectively). And *pex16¹* brain compared to rescue (ratio 0.249, p=0.002), and for *pex16^{EY}* compared to rescue (ratio 0.756, p<0.001). **B.** Levels in mol% of PE 33:2 in *pex2* and *pex16* mutant brains show significant decreases in *pex* mutant brains. For PE 33:2 in *pex2²* brain and *pex2¹* brain, levels are dramatically reduced compared to controls (ratio 0.403, 0.469 respectively, p<0.001, p<0.001 respectively) and *pex2²* brain compared to rescue (ratio 0.430, p<0.001), and *pex2¹* compared to rescue (ratio 0.496, p<0.001). For PE 33:2 in *pex16¹* and *pex16^{EY}* brain, levels are decreased compared to controls (ratio 0.475, 0.688 respectively,

p<0.001, p<0.001 respectively). And *pex16¹* brain compared to rescue (ratio 0.494, p=0.006), and for *pex16^{EY}* compared to rescue (ratio 0.658, p=0.001). **C.** Levels in mol% of PE 33:1 in *pex2* and *pex16* mutant brains show significant decreases in *pex* mutant brains. For PE 33:3 in *pex2²* brain and *pex2¹* brain, levels are dramatically reduced compared to controls (ratio 0.549, 0.578 respectively, p<0.001, p<0.001 respectively) and *pex2²* brain compared to rescue (ratio 0.528, p<0.001), and *pex2¹* compared to rescue (ratio 0.510, p<0.001). For PE 33:3 in *pex16¹* levels are decreased compared to controls (ratio 0.789, p<0.001). And *pex16¹* brain compared to rescue (ratio 0.758, p=0.019). For the *pex16^{EY}* the difference was not significant from control, but was reduced compared to rescue (ratio 0.868, p=0.006). **D.** Levels in mol% of PE 35:4 in *pex2* and *pex16* mutant brains show significant decreases in *pex* mutant brains. For PE 35:4 in *pex2²* brain and *pex2¹* brain, levels are dramatically reduced compared to controls (ratio 0.400, 0.467 respectively, p<0.001, p<0.001 respectively) and *pex2²* brain compared to rescue (ratio 0.380, p<0.001), and *pex2¹* compared to rescue (ratio 0.528, p<0.001). For PE 35:4 in *pex16¹* levels are decreased compared to controls (ratio 0.443, p=0.002). And *pex16¹* brain compared to rescue (ratio 0.433, p=0.047). For the *pex16^{EY}* the difference was not significant from control, but was reduced compared to rescue (ratio 0.749, p<0.001). **E.** Levels in mol% of PE 35:3 in *pex2* and *pex16* mutant brains show significant decreases in *pex* mutant brains. For PE 35:3 in *pex2²* brain and *pex2¹* brain, levels are dramatically reduced compared to controls (ratio 0.131, 0.119 respectively, p<0.001, p<0.001 respectively) and *pex2²* brain compared to rescue (ratio 0.110, p<0.001), and *pex2¹* compared to rescue (ratio 0.101, p<0.001). For PE 35:3 in *pex16¹* and *pex16^{EY}* levels are decreased compared to controls (ratio 0.141, 0.644, p<0.001, P<0.001) and for *pex16¹* brain compared to rescue (ratio 0.181, p=0.001) and for *pex16^{EY}* compared to rescue (ratio 0.677, p<0.001). **F.** Levels in mol% of PE 35:2 in *pex2* and *pex16* mutant brains show significant decreases in *pex* mutant brains. For PE 35:2 in *pex2²* brain and *pex2¹* brain, levels are dramatically reduced compared to controls (ratio 0.118, 0.124, respectively, p < 0.001, p < 0.001 respectively) and *pex2²* brain compared to rescue (ratio 0.104, p < 0.001), and *pex2¹* compared to rescue (ratio 0.115, p < 0.001). For PE 35:2 in *pex16¹* and *pex16^{EY}* levels are decreased compared to controls (ratio 0.186, 0.540, p < 0.001, P < 0.001) and for *pex16¹* brain compared to rescue (ratio 0.227, p = 0.006) and for *pex16^{EY}* compared to rescue (ratio 0.527, p < 0.001). (TIF)

**Fig S7. Robust chain-length specific elevations in some intermediate chain and long chain PE phospholipids in peroxisomal mutant brain. A.** Levels in mol% of PE 33:0 in *pex2* and *pex16* mutant brains show significant increases in *pex* mutant brains. For PE 33:0 in *pex2²* brain and *pex2¹* brain, levels are increased compared to controls (ratio 2.227, 2.153 respectively, p < 0.001, p = 0.009 respectively) and *pex2²* brain compared to rescue (ratio 2.032, p < 0.001), and *pex2¹* compared to rescue (ratio 1.644, p = 0.044). For PE 33:0 in *pex16¹* and *pex16^{EY}* brain, levels are increased compared to controls (ratio 1.834, 1.900 respectively, p = 0.005, p = 0.001 respectively) and for *pex16^{EY}* compared to rescue (ratio 1.518, p = 0.003). **B.** Levels in mol% of PE 34:4 in *pex2* and *pex16* mutant brains show significant decreases in *pex* mutant brains. For PE 34:4 in *pex2²* brain and *pex2¹* brain, levels are increased compared to controls (ratio 1.285, 1.202 respectively, p=0.001, p=0.005 respectively) and *pex2²* brain compared to rescue (ratio 1.151, p=0.002). For PE 34:4 in *pex16¹* and *pex16^{EY}* brain, levels are increased compared to controls (ratio 1.366, 1.136 respectively, p<0.001, p=0.048 respectively) but no differences from rescue. **C.** Levels in mol% of PE 34:3 in *pex2* and *pex16* mutant brains show significant increases in *pex* mutant brains. For PE 34:3 in *pex2²* brain and *pex2¹* brain, levels are increased compared to controls (ratio 1.327, 1.343 respectively, p<0.001, p<0.001 respectively) and *pex2²* brain compared to rescue (ratio 1.170, p<0.001), and *pex2¹* compared to rescue (ratio 1.094, p=0.002). For PE 34:3 in *pex16¹* and *pex16^{EY}* brain, levels are increased compared to controls (ratio 1.388, 1.157 respectively, p<0.001, p<0.001 respectively) and for *pex16¹* compared to rescue (ratio 1.141, p=0.003). **D.** Levels in mol% of PE 34:1 in *pex2* and *pex16* mutant brains show significant increases in *pex* mutant brains. For PE 34:1 in *pex2²* brain and *pex2¹* brain, levels are increased compared to controls (ratio 1.371, 1.328 respectively, p<0.001, p<0.001 respectively) and *pex2²* brain compared to rescue (ratio 1.219, p<0.001), and *pex2¹* compared to rescue (ratio 1.141, p<0.001). For PE 34:1 in *pex16¹* and *pex16^{EY}* brain, levels

are increased compared to controls (ratio 1.249, 1.140 respectively, p<0.001, p<0.001 respectively) and for *pex16^EY* compared to rescue (ratio 1.076, p<0.001). **E.** Levels in mol% of PE 36:3 in *pex2* and *pex16* mutant brains show significant increases in *pex* mutant brains. For PE 36:3 in *pex2²* brain and *pex2¹* brain, levels are increased compared to controls (ratio 1.180, 1.240 respectively, p<0.001, p<0.001 respectively) and *pex2²* brain compared to rescue (ratio 1.056, p=0.025), and *pex2¹* compared to rescue (ratio 1.068, p=0.009). For PE 36:3 in *pex16¹* and *pex16^EY* brain, levels are increased compared to controls (ratio 1.112, 1.157 respectively, p<0.001, p<0.001 respectively) and for *pex16^EY* compared to rescue (ratio 1.092, p=0.002). **F.** Levels in mol% of PE 38:3 in *pex2* and *pex16* mutant brains show significant increases in *pex* mutant brains. For PE 38:3 in *pex2²* brain and *pex2¹* brain, levels are increased compared to controls (ratio 1.795, 1.717 respectively, p<0.001, p<0.001 respectively) and *pex2²* brain compared to rescue (ratio 1.661, p<0.001), and *pex2¹* compared to rescue (ratio 1.540, p<0.001). For PE 38:3 in *pex16¹* and *pex16^EY* brain, levels are increased compared to controls (ratio 1.393, 1.382 respectively, p<0.001, p=0.001 respectively) and for *pex16¹* compared to rescue (ratio 1.201, p=0.005) *pex16^EY* compared to rescue (ratio 1.157, p=0.029).
(TIF)

**Fig S8. Lyso-Phospholipid abnormalities in human plasma from patients with PBD-ZSD. A.** Human Plasma LPC(16:0) is decreased in patients with PEX1 mutations compared to pediatric (ratio 0.523, p<0.001) and adult controls (ratio 0.656, p<0.001). **B.** Human Plasma LPC(18:0) is decreased in patients with PEX1 mutations compared to pediatric (ratio 0.467, p<0.001) and adult controls (ratio 0.620, p=0.001).
(TIF)

**Table S1. Acquisition and data processing parameters for human plasma using a Waters Xevo TQS triple quadrupole mass spectrometer.**
(XLSX)

**Table S2. Lipid composition of Drosophila pex mutants (third instar) as analyzed by direct infusion electrospray ionization mass spectrometry, using precursor and neutral loss scans specific for each lipid class or group.** Polar lipids are presented as percent of total polar lipids in each sample. Analyzed DAGs, which include those containing the common Drosophila fatty acids 16:1 and 18:1 are presented as the percent of normalized DAG intensity detected in each sample, and TAGs are similarly presented as percent of normalized TAG intensity detected in each sample. Lipid names in column A indicate total acyl carbons:total acyl double bonds. DAG names in column B indicate each acyl chain; TAG names in column B indicate one acyl chain and the other two chains combined.
(XLSX)

**Table S3. Lipid composition of heads of adult Drosophila pex mutants as analyzed by direct infusion electrospray ionization mass spectrometry, using precursor and neutral loss scans specific for each lipid class or group.** Polar lipids are presented as percent of total polar lipids in each sample. Analyzed DAGs, which include those containing the common Droshphila fatty acids 16:1 and 18:1 are presented as the percent of normalized DAG intensity detected in each sample, and TAGs are similarly presented as percent of normalized TAG intensity detected in each sample. Lipid names in column A indicate total acyl carbons:total acyl double bonds. DAG names in column B indicate each acyl chain; TAG names in column B indicate one acyl chain and the other two chains combined.
(XLSX)

**Table S4. Lipid composition of human plasma from subject with PEX variations, as analyzed by direct infusion electrospray ionization mass spectrometry on a Waters Xevo TQS mass spectrometer, using precursor and neutral loss scans specific for each lipid class or group.** Lipids are presented as percent of normalized (to internal standards) mass spectral signal in total polar lipids in each sample. Quality control samples were made by pooling aliquots

from each sample. Quality control samples (n = 7) were run at regular intervals among the other samples and were used to indicate the quality of the analysis for each lipid analyte, using criteria of limit of detection (0.0005 nmol, column F) and coefficient of variation (standard deviation/average of quality control samples, column G).
(XLSX)

## Author contributions

**Conceptualization:** Michael Francis Wangler, Ruth Welti, James A. McNew.

**Data curation:** Yu-Hsin Chao, Mary Roth, Ruth Welti.

**Formal analysis:** Michael Francis Wangler, Yu-Hsin Chao, Mary Roth, Ruth Welti.

**Funding acquisition:** Michael Francis Wangler, James A. McNew.

**Investigation:** Michael Francis Wangler, Mary Roth, James A. McNew.

**Methodology:** Michael Francis Wangler, Mary Roth.

**Project administration:** Michael Francis Wangler.

**Supervision:** Michael Francis Wangler.

**Visualization:** Michael Francis Wangler.

**Writing – original draft:** Michael Francis Wangler.

**Writing – review & editing:** Yu-Hsin Chao, Mary Roth, Ruth Welti, James A. McNew.

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
