## [Decision Letter · Decision Letter 0]

23 Sep 2024

Dear Dr. Wangler,

Thank you for submitting your manuscript to PLOS ONE. After careful consideration, we feel that it has merit but does not fully meet PLOS ONE’s publication criteria as it currently stands. Therefore, we invite you to submit a revised version of the manuscript that addresses the points raised during the review process.

We look forward to receiving your revised manuscript.

Kind regards,

Juan J Loor

Academic Editor

PLOS ONE

2. Thank you for stating the following financial disclosure: [This work was supported by the National Institute for Neurological Disorders and Stroke 5R01NS107733 to MFW. The lipid analyses described in this work were performed at the Kansas Lipidomics Research Center Analytical Laboratory. Instrument acquisition and lipidomics method development were supported by the National Science Foundation (including support from the Major Research Instrumentation program; most recent award DBI-1726527), K-IDeA Networks of Biomedical Research Excellence (INBRE) of National Institute of Health (P20GM103418), USDA National Institute of Food and Agriculture (Hatch/Multi-State project 1013013), and Kansas State University.]. Please state what role the funders took in the study. If the funders had no role, please state: "The funders had no role in study design, data collection and analysis, decision to publish, or preparation of the manuscript." If this statement is not correct you must amend it as needed. Please include this amended Role of Funder statement in your cover letter; we will change the online submission form on your behalf.

Additional Editor Comments (if provided):

Reviewers' comments:

Reviewer's Responses to Questions

**Comments to the Author**

1. Is the manuscript technically sound, and do the data support the conclusions?

Reviewer #1: Yes

Reviewer #2: Yes

Reviewer #3: Yes

2. Has the statistical analysis been performed appropriately and rigorously?

Reviewer #1: Yes

Reviewer #2: No

Reviewer #3: Yes

3. Have the authors made all data underlying the findings in their manuscript fully available?

Reviewer #1: Yes

Reviewer #2: Yes

Reviewer #3: Yes

4. Is the manuscript presented in an intelligible fashion and written in standard English?

Reviewer #1: Yes

Reviewer #2: Yes

Reviewer #3: Yes

Reviewer #1: Although increased levels of very long chain fatty acids have been known to be hallmarks of peroxisome biogenesis disorders, the authors and others previously discovered that other lipid metabolites are dysregulated. The authors now further investigated glycerophospholipid and sphingolipid changes in Drosophila models and in human plasma and attempted to deduce mechanisms underlying the abnormalities.

The experimental work was done thoroughly and the data were in general extensively described and illustrated.

In the paragraph on changes in lipid composition in human plasma the data are not always accurately described and they raise several questions. This paragraph should be thoroughly revised. Does the title reflect the data? I do not understand based on the data in Fig 8 D and E how the huge difference in the ratio SM C24/22 can be explained. Also, how do the authors explain the reduced levels of SM C22 in combination with normal levels of SM C24? In several samples the SM C22 seem not to be detectable. How were calculations made and how was this handled in the statistics?

“We also observed a decrease in the total levels of plasma ether-linked PC, i.e., alkyl or plasmenyl PC, here referred to as “ePC”.” This is to be expected in a PBD but at least a short explanation should be given for the non-expert reader. “Chain-length abnormalities were also observed, including decreased levels of PC 34:2 (Figure 6C), PC(36:2) (Figure 6D), PC 38:2 (Figure 6E), and PC 38:1 (Figure 6F) with an increase in the long chain PC 38:1 in patients with PEX1 mutations compared to controls (Figure 6B).” Why is PC38:1 mentioned twice and in a contradictory way? Also take care of the writing of chain lengths (with or without parantheses). “Plasma DSM 22:0 was increased (Figure 7E)”: this does not correspond with the figure. DSM22 was not defined.

“The excess levels of very long chain lipids in peroxisomal mutants lead to overabundance of long chain sphingolipids”: the authors should discuss how excess VLCFA can impact on long chain FA.

Minor remarks

Introduction: “Our results suggest that peroxisomal biogenesis defects alter shuttling of the acyl chains of multiple phospholipid and ceramide lipid classes, whereas DAG species with intermediate fatty acids are more abundant”- the two parts in this sentence do not seem to fit together.

Changes in levels of ether lipids were mentioned in the results on human plasma but the relevance and impact were not discussed

Discussion: “However, it can reveal evidence for shuttling as 1) peroxisomes are required for very long chain fatty acid breakdown 2) increased levels of C24 and C26 derivatives of many biochemical lipids have been previously observed and 3) it seems likely that that is the proximal cause of the phospholipid alterations, and the changes observed in Intermediate chain lipids are the primary cause.” This sentence is not clear (what is the difference between proximal and primary?) and should be improved.

I would include the most comprehensive recent review on peroxisomal function which was written by Wanders et al in the references (Physiol Rev, 2023).

Reviewer #2: This manuscript described the alterations in phospholipids and diacylglycerol in Peroxisomal Biogenesis Disorders using Pex2 and Pex16 mutant drosophila model and human plasma samples. Although the lipidomics data revealed the significant changes in several lipid species, the authors did not provide any information on statistical analysis. Furthermore, they presented ratio changes in several figures, such as Fig. 3,4,5. Whether these ratio changes are significant or not was not clear. They need to provide p values for significant changes in these graphs.

Reviewer #3: In this paper entitled, “Drosophila Models Uncover Substrate Channeling Effects on Phospholipids and Sphingolipids in Peroxisomal Biogenesis Disorders” by Wangler et al reports on lipid fluctuation of Drosophila larvae and adult heads with peroxisomal deficiencies. They used pex2 and pex16 mutant flies for their analysis and found that phospholipids such as phosphatidylcholine (PC) and phosphatidylethanolamine (PE) show specific patterns of increase and decrease in certain phospholipid species, mainly according to carbon chain length. They also found that diacylglycerol shows an inverse fluctuation pattern compared to phospholipids. The manuscript is well written and contains valuable information in this research field. Below are my comments on the manuscript before publication.

Accumulation of very long chain fatty acid (VLCFA) is a characteristic feature of patients with PBD-ZSD. In the author's experiments, it is not clear whether PE and PC containing VLCFA are increased. Please explain about this (for example whether they are detected or not). Same for DAGs.

The process of lipid species identification and quantification by direct infusion MS is unclear. Could authors please explain in detail how lipids are identified, acyl chains are determined, and quantified? An illustration would be helpful.

The method of lipid preparation form human plasma and Drosophila samples are missing. I suggest including them in the method section.

Can the authors also show the overall lipid species identified in the human plasma samples as shown for the Drosophila samples in Supplementary Table 2?

Minor comments:

In method section: “Water Xevo TQS” would be “Waters Xevo TQS”

In method section, Fly husbandry: The fly genotypes shown are somewhat confusing. Please separate each genotype with appropriate commas and periods to distinguish the different genotypes.

In Figure1: As same for above, the genotypes used should be shown more explicitly (especially for adult flies). Please separate each genotype clearly.

In Figure5: Color coding for fly genotypes is not easily distinguishable.

**Do you want your identity to be public for this peer review?** For information about this choice, including consent withdrawal, please see our Privacy Policy

Reviewer #1: No

Reviewer #2: **Yes: ** Bo Wang

Reviewer #3: No

---

## [Author Response · Author response to Decision Letter 1]

23 Feb 2025

Reviewer #1: Although increased levels of very long chain fatty acids have been known to be hallmarks of peroxisome biogenesis disorders, the authors and others previously discovered that other lipid metabolites are dysregulated. The authors now further investigated glycerophospholipid and sphingolipid changes in Drosophila models and in human plasma and attempted to deduce mechanisms underlying the abnormalities.

The experimental work was done thoroughly and the data were in general extensively described and illustrated.

We thank the reviewer for the assessment of our work and we agree, looking beyond the most proximal peroxisomal biochemical changes out into the lipidomic profile is an essential step in understanding peroxisomal disorders.

In the paragraph on changes in lipid composition in human plasma the data are not always accurately described and they raise several questions. This paragraph should be thoroughly revised.

According to the reviewers suggestion we have revised this paragraph to more accurately discuss the findings of the 2016 Herzog publication to specify the fibroblast findings and the identification of phospholipids that contain a very long chain fatty acid. We have removed a reference to sphingomyelin in this same paper and we have discussed the potential for novel biomarkers as noted in Herzog et al 2018. See Page 3, second paragraph of the introduction.

Does the title reflect the data?

The best explanation for all the data we have generated is that there is a substrate shuttling effect that leads to more rapid or prevalent remodeling of phospholipids by the Land’s cycle versus de novo synthesis in the Kennedy pathway (Figure S1) and results in a relative decrease in short and intermediate chains compared to long chains. This is supported by all the data presented from Drosophila larvae that gave us the original hypothesis and demonstrated the imbalance between DAG precursors and PE and PC products for intermediate chains and long chains (Figures 2-3, Figures S2-S4), it explains data we published in previous publications from patients with PBD showing a decrease in intermediate chain SMs (Wangler et al. 2018), it was supported by additional data from Drosophila adult brain (Figure 4-5, Figures S5-S9) and a new human plasma study (Figure 6-7, Figure S10-S11) conducted on a very rare disorder with two control groups. If substrate shuttling was not present then the imbalances in the DAG precursors would feed forward (according to Figure S1) and correct the defects that we consistently see in both fly models and disease.

I do not understand based on the data in Fig 8 D and E how the huge difference in the ratio SM C24/22 can be explained.

Also, how do the authors explain the reduced levels of SM C22 in combination with normal levels of SM C24? In several samples the SM C22 seem not to be detectable. How were calculations made and how was this handled in the statistics?

We assume this refers to Figure 7F. The PBD plasma samples clearly show in Figure 7A-D a pattern of reduced SMs with the exception of SM(24:0). This is the fundamental point of this analysis. We agree with the reviewer that the ratio is subject to extreme variability because of such low levels of SM C22 in the PBD patient samples. We have therefore removed Figure 7E and F. We thank the reviewer for pointing out this problem.

Note that in Discussion we also state: “we cannot rule out an overall deficiency of sphingomyelins with a disproportionate quantity of the remaining having long or very long chain acyl chains.” Page 9

“We also observed a decrease in the total levels of plasma ether-linked PC, i.e., alkyl or plasmenyl PC, here referred to as “ePC”.” This is to be expected in a PBD but at least a short explanation should be given for the non-expert reader.

We appreciate the suggestion, please see brief comment in results page 7.

“Chain-length abnormalities were also observed, including decreased levels of PC 34:2 (Figure 6C), PC(36:2) (Figure 6D), PC 38:2 (Figure 6E), and PC 38:1 (Figure 6F) with an increase in the long chain PC 38:1 in patients with PEX1 mutations compared to controls (Figure 6B).” Why is PC38:1 mentioned twice and in a contradictory way? Also take care of the writing of chain lengths (with or without parantheses). “Plasma DSM 22:0 was increased (Figure 7E)”: this does not correspond with the figure. DSM22 was not defined.

We appreciate the reviewer for pointing out these errors. The PC38:1 and PC38:2 levels are increased, this has been corrected, please see results page 7.

“The excess levels of very long chain lipids in peroxisomal mutants lead to overabundance of long chain sphingolipids”: the authors should discuss how excess VLCFA can impact on long chain FA.

We appreciate the reviewer for pointing out this error. We have added the word “very” to this sentence (see page 9).

Minor remarks

Introduction: “Our results suggest that peroxisomal biogenesis defects alter shuttling of the acyl chains of multiple phospholipid and ceramide lipid classes, whereas DAG species with intermediate fatty acids are more abundant”- the two parts in this sentence do not seem to fit together.

We have made this into two sentences, see page 2.

Changes in levels of ether lipids were mentioned in the results on human plasma but the relevance and impact were not discussed.

We have taken the reviewers suggestion and added a brief comment on the plasmalogen impact in results.

Discussion: “However, it can reveal evidence for shuttling as 1) peroxisomes are required for very long chain fatty acid breakdown 2) increased levels of C24 and C26 derivatives of many biochemical lipids have been previously observed and 3) it seems likely that that is the proximal cause of the phospholipid alterations, and the changes observed in Intermediate chain lipids are the primary cause.” This sentence is not clear (what is the difference between proximal and primary?) and should be improved.

We thank the reviewer, we have made this paragraph more clear, please see the revision, page 8 “We propose a model…”

I would include the most comprehensive recent review on peroxisomal function which was written by Wanders et al in the references (Physiol Rev, 2023).

We thank the reviewer we have added this reference.

Reviewer #2: This manuscript described the alterations in phospholipids and diacylglycerol in Peroxisomal Biogenesis Disorders using Pex2 and Pex16 mutant drosophila model and human plasma samples. Although the lipidomics data revealed the significant changes in several lipid species, the authors did not provide any information on statistical analysis.

Furthermore, they presented ratio changes in several figures, such as Fig. 3,4,5. Whether these ratio changes are significant or not was not clear. They need to provide p values for significant changes in these graphs.

We thank the reviewer for these thoughtful comments. We have added a statistical comparisons statement to the Methods section and each figure legend contains the ratio and the significance between mutant and control. We have removed the ratios in favor of lipid levels in Figure 4,5 and removed Figure S9. For Figure 3, the ratios are point estimates of the average mutants over the average rescue and the biological trend is the main finding, and the significance for lipid classes is shown in Figure 2. Moreover, putting statistical stars above the ratio graphs would be overwhelming and illegible, Figure 3A has 38 lipid classes on the X axis and two ratios per class. We have provided all the data values in the supplemental tables for all the compounds in the entire lipidomic analysis.

Reviewer #3: In this paper entitled, “Drosophila Models Uncover Substrate Channeling Effects on Phospholipids and Sphingolipids in Peroxisomal Biogenesis Disorders” by Wangler et al reports on lipid fluctuation of Drosophila larvae and adult heads with peroxisomal deficiencies. They used pex2 and pex16 mutant flies for their analysis and found that phospholipids such as phosphatidylcholine (PC) and phosphatidylethanolamine (PE) show specific patterns of increase and decrease in certain phospholipid species, mainly according to carbon chain length. They also found that diacylglycerol shows an inverse fluctuation pattern compared to phospholipids. The manuscript is well written and contains valuable information in this research field. Below are my comments on the manuscript before publication.

Accumulation of very long chain fatty acid (VLCFA) is a characteristic feature of patients with PBD-ZSD. In the author's experiments, it is not clear whether PE and PC containing VLCFA are increased. Please explain about this (for example whether they are detected or not). Same for DAGs.

Thanks for this comment. The PC and PEs have two acyl side chains and the mass spec data detects the total carbons for the two side chains in sum. We direct the reviewer to our explanation of the mass spec detection of these lipid classes on page 5 where we state: “Note that in this nomenclature for PC and PE, the 30:1 means that the two acyl chains have a total of 30 carbons and 1 unsaturation total. Across the spectrum of phospholipids measured, all the acyl groups are likely between C12 and C22 and would be considered long chain fatty acids or acyl groups. However, we observed a pattern specific to 28 and 30 carbon that differed from 38 carbon, for example. In order to characterize this, we distinguish between broad classes of long chain phospholipids and so we term a phospholipid to have an intermediate chain length if the total carbon chain (across the two acyl chains) is C28-C31, and we consider C32 and above to be long chain length phospholipids. Indeed, a number of intermediate chain length PC and PE’s exhibit the same pattern. PC 28:1 and PC 28:0 are both reduced in pex2 mutants compared to rescue and pex16 mutant compared to rescue (Figure S2A).”

The process of lipid species identification and quantification by direct infusion MS is unclear. Could authors please explain in detail how lipids are identified, acyl chains are determined, and quantified? An illustration would be helpful.

The specific acyl chains are not identified, but the sum of the acyl chains and unsaturations for the two acyl chains in PC and PE. The paper does not designate specific acyl chains in the results section. However, inferences can be made in the discussion given the well known role of peroxisomes in the metabolism of VLCFA.

The method of lipid preparation form human plasma and Drosophila samples are missing. I suggest including them in the method section.

We thank the reviewer and have added a Lipid Extraction section to Materials and Methods.

Can the authors also show the overall lipid species identified in the human plasma samples as shown for the Drosophila samples in Supplementary Table 2?

This data is indeed provided in Supplementary Table 4

Minor comments:

In method section: “Water Xevo TQS” would be “Waters Xevo TQS”

This has been corrected

In method section, Fly husbandry: The fly genotypes shown are somewhat confusing. Please separate each genotype with appropriate commas and periods to distinguish the different genotypes.

We have put the genotypes on separate lines for easier review.

In Figure1: As same for above, the genotypes used should be shown more explicitly (especially for adult flies). Please separate each genotype clearly.

Putting full genotypes in the Table would be illegible, and we have specifically noted in Methods which genotypes e have labeled and how that corresponds to each of these designations. Moreover, the graphs later in the paper use these same designations. We have added the following statement for Figure 1C legend “The boxes show labels for these lines, for full genotypes please refer to the methods section.”

In Figure5: Color coding for fly genotypes is not easily distinguishable.

We removed the Ratio figures in favor of clarity and simplicity in Figure 5.

---

## [Decision Letter · Decision Letter 1]

22 Apr 2025

Drosophila Models Uncover Substrate Channeling Effects on Phospholipids and Sphingolipids in Peroxisomal Biogenesis Disorders

PONE-D-24-26309R1

Dear Dr. Wangler,

We’re pleased to inform you that your manuscript has been judged scientifically suitable for publication and will be formally accepted for publication once it meets all outstanding technical requirements.

Kind regards,

Juan J Loor

Academic Editor

PLOS ONE

Additional Editor Comments (optional):

Reviewers' comments:

Reviewer's Responses to Questions

**Comments to the Author**

Reviewer #1: All comments have been addressed

Reviewer #2: All comments have been addressed

2. Is the manuscript technically sound, and do the data support the conclusions?

Reviewer #1: Yes

Reviewer #2: Yes

3. Has the statistical analysis been performed appropriately and rigorously?

Reviewer #1: Yes

Reviewer #2: Yes

4. Have the authors made all data underlying the findings in their manuscript fully available?

Reviewer #1: Yes

Reviewer #2: Yes

5. Is the manuscript presented in an intelligible fashion and written in standard English?

Reviewer #1: Yes

Reviewer #2: Yes

Reviewer #1: (No Response)

Reviewer #2: (No Response)

**Do you want your identity to be public for this peer review?** For information about this choice, including consent withdrawal, please see our Privacy Policy

Reviewer #1: No

Reviewer #2: No

---

## [Editor Report · Acceptance letter]

PONE-D-24-26309R1

PLOS ONE

Dear Dr. Wangler,

I'm pleased to inform you that your manuscript has been deemed suitable for publication in PLOS ONE. Congratulations! Your manuscript is now being handed over to our production team.

Kind regards,

on behalf of

Dr. Juan J Loor

Academic Editor

PLOS ONE